# Task-Awareness Improves LLM Generations and Uncertainty

**Tim Tomov** [* 1 2 3]   **Dominik Fuchsgruber** [* 1 2]   **Stephan Günnemann** [1 2 3]

## Abstract

In many applications of LLMs, natural language responses often have an underlying structure such as representing discrete labels, numerical values, or graphs. Yet, existing decoding and uncertainty estimation methods operate only in language space and largely disregard structural information. We address this by modeling LLM outputs directly in a task-dependent latent structure. By equipping this structure with a dissimilarity measure, we can compute Bayes-optimal responses. These are not selected from sampled generations but are newly synthesized by combining individual responses in the latent space. Across different tasks, Bayes-optimal responses consistently outperform standard decoding methods like beam search. Moreover, quantifying uncertainty via the induced Bayesian risk captures variations in terms of the latent structure and improves alignment with output quality and correctness. Our decision-theoretic framework is applicable to any problem that admits a latent response structure and enables reliable task-aware LLM predictions.

## 1. Introduction

Large language models (LLMs) have emerged as highly general-purpose systems, successfully generalizing across tasks ranging from factual question answering (Kuhn et al., 2023; Duan et al., 2024) to machine translation (Vilar et al., 2023), and text summarization (Zhang et al., 2024), among many others (Raiaan et al., 2024; Hadi et al., 2023). When prompting LLMs, the semantics of the response are unknown a priori, but the domain of the expected output is often implicitly given. For example, when asking "What is the capital of Namibia?", the output should be a city. Simi-

larly, when asking an LLM to rate an essay on a scale from 0 to 10, one expects the output to be a number within this range. While LLMs are trained to generate responses in natural language, the task itself is often associated with a clear task-specific *latent structure*.

Compared to free-form natural language, these latent structures are typically better suited for interpreting and comparing LLM predictions in downstream settings. In many automated workflows, decisions depend on the task-level semantics rather than the linguistic properties. Consequently, LLM outputs are commonly post-processed to recover a task-specific representation such as a numeric score or categorical label (Mekala et al., 2023; Zhong et al., 2024; Lukasik et al., 2024). In short: *The latent representations of LLM responses are often more useful than the text itself.*

We leverage this insight to improve the generation quality of LLMs and the corresponding estimates of their uncertainty. To that end, we introduce a highly general framework which can be used for any task that admits a latent response structure. Defining a mapping from natural language to the task-specific latent representation induces an LLM's predictive distribution in the latent space. Utilizing a task-dependent notion of similarity in the latent structure, we perform Minimum Bayes Risk (MBR) decoding (Kumar & Byrne, 2004) to obtain a *Bayes-optimal* response *in the latent space*. This response needs not to correspond to any of sampled generations and can instead be synthesized by aggregating information across the model's full output distribution. This is in general not possible in the domain of raw language responses. At the same time, variability in the latent space with respect to the task-specific dissimilarity yields an uncertainty estimate that directly reflects the notion of error relevant to the task, naturally quantified as *Bayes risk*.

We demonstrate the effectiveness of our framework over a broad range of tasks: Single-answer and multi-answer question answering, text summarization, and machine translation. Across all settings, Bayes-optimal latent responses consistently outperform other decoding schemes, such as beam search or self-consistency (Wang et al., 2023). At the same time, the Bayes risk provides uncertainty estimates that are more informative than existing approaches in terms of the correctness and quality of the corresponding responses.

[1]School of Computation, Information & Technology, Technical University of Munich [2]Munich Data Science Institute [3]Munich Center for Machine Learning. Correspondence to: Tim Tomov <tim.tomov@tum.de>, Dominik Fuchsgruber <d.fuchsgruber@tum.de>.

*Proceedings of the 43rd International Conference on Machine Learning*, Seoul, South Korea. PMLR 306, 2026. Copyright 2026 by the author(s).

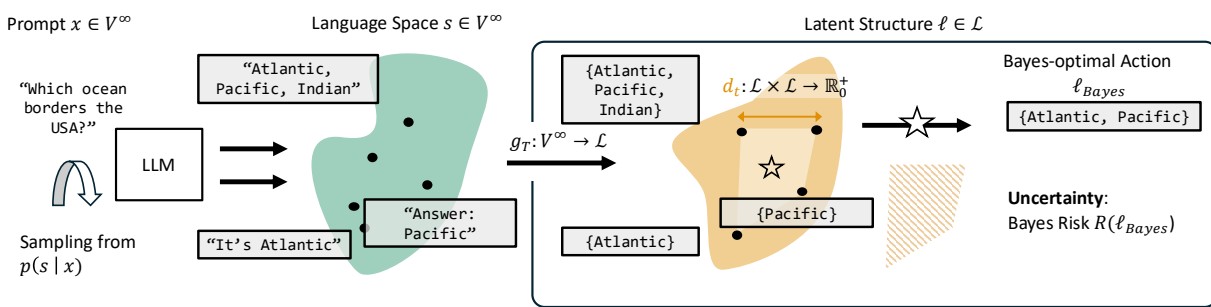

*Figure 1.* Framework of embedding LLM responses $s \mid x$ into a task-dependent latent space $\mathcal{L}$ on the example of set-based multi-answer question answering. The Bayes-optimal $\ell_{\text{Bayes}}$ response is the centroid in the latent space w.r.t. a distance metric $d_T$. It does not need to be generated by the LLM directly. Uncertainty is quantified as Bayesian risk that measures the spread in $p(\ell \mid x)$ w.r.t. to $d_T$.

## 2. Latent Structures Enable Better LLM Generations and Uncertainty Quantification

LLMs implicitly define a probability distribution over arbitrary-length strings $s \in \mathcal{V}^\infty$ from a vocabulary $\mathcal{V}$, which we denote by $p(s)$. Conditioning on an input prompt $x \in \mathcal{V}^\infty$ yields the predictive distribution $p(s \mid x)$. In downstream applications, these responses inherently admit a task-specific representation beyond *free-form language*. For example, problems like question answering (QA) restrict the LLM's outputs to a discrete support set. In rating problems answers are embedded as reals or ordinals. Treating LLM responses as latents in some structure also has the benefit of enabling further application-specific processing.

Formally, let $T$ be a task with an associated latent space $\mathcal{L}$. We define an encoding function $g_T : \mathcal{V}^\infty \to \mathcal{L}$ that projects any generated string $s$ onto its task-dependent latent representation $\ell$ (e.g., a real number, a categorical class label). Figure 1 shows an example from multi-answer QA where $g_T$ maps raw strings to a set of possible answers. The mapping $g_T$ then induces a push-forward distribution of the LLM's responses over latent responses $\ell$

$$p(\ell \mid x) \;=\; \sum_{s \in \mathcal{V}^\infty} p(s \mid x)\, \mathbf{1}[g_T(s) = \ell]. \quad (1)$$

To compare the generations $s$ in the latent space $\mathcal{L}$, we equip $\mathcal{L}$ with an appropriate dissimilarity measure. In principle, any measure $d : \mathcal{L} \times \mathcal{L} \to \mathbb{R}_{\geq 0}$ is possible, but instantiating $d$ with a proper metric has desirable properties regarding uncertainty estimation as we will detail in Section 2.2. Any particular choice of $d$ encodes a task-specific notion of error, for example, the Hamming distance, F1 score, or 0-1 loss.

### 2.1. Structure-Aware Minimum Bayes Risk Decoding

We directly leverage this structure-dependent representation to synthesize a latent response that does not directly correspond to any of the LLM's generations. To that end, we utilize the distribution $p(\ell \mid x)$ induced by mapping $g_T$ through Bayesian decision theory. For any candidate action

(LLM response) $\tilde{\ell} \in \mathcal{L}$, the associated *Bayes risk* under the dissimilarity $d_T$ can be defined as

$$R(\tilde{\ell}) \;:=\; \mathbb{E}_{\ell \sim p(\ell|x)}\Big[d_T(\tilde{\ell}, \ell)\Big]. \quad (2)$$

The *Bayes-optimal action* (LLM response) is given by

$$\ell_{\text{Bayes}} \;:=\; \arg\min_{\tilde{\ell} \in \mathcal{L}} R(\tilde{\ell}), \quad (3)$$

with the associated *Bayes risk*

$$R_{\text{Bayes}} := R(\ell_{\text{Bayes}}). \quad (4)$$

Selecting a response that minimizes this risk is referred to as Minimum Bayes Risk (MBR) decoding (Kumar & Byrne, 2004): $\ell_{\text{Bayes}}$ represents the optimal action with respect to minimizing Bayes risk under the response distribution $p$. While MBR has been explored using different dissimilarity functions $d$ (Bertsch et al., 2023; Freitag et al., 2023; Eikema et al., 2025) in the space of raw language, we are the first to apply MBR directly in the latent space $\mathcal{L}$.

A key property is that for many latent structures and dissimilarities the Bayes-optimal action (Equation (3)) has a closed-form expression or can be computed efficiently. At the same time, it is infeasible to compute the Bayes-optimal closed-form raw language answer as there is no tractable framework for aggregating over raw language. E.g., we can not compute the intersection of two textual responses $x$, but their latent representations as sets of discrete answers permit such operations at negligible cost.

**Approximating the Distribution over Latent Responses**
In practice, we do not have direct access to $p(\ell \mid x)$ and must approximate it via $M$ Monte-Carlo samples $\{s^{(i)}\}_{i=1}^M$:

$$p(\ell \mid x) \approx \frac{1}{M} \sum_{i=1}^M \mathbf{1}[g_T(s^{(i)}) = \ell]. \quad (5)$$

Operating in language space, existing MBR approaches cannot do better than selecting the LLM generation associated

with the lowest risk

$$\ell_{\text{Sample}} = \arg \min_{\ell^{(i)}} R(\ell^{(i)}) \qquad (6)$$

This optimization requires $O(M^2)$ pairwise comparisons between the generated responses. For many structures, computing the optimal response $\ell_{\text{Bayes}}$ only has computational complexity of $O(M)$, compare Table 1. Importantly, by leveraging the latent structure $\mathcal{L}$, we can compute the optimal response *without the LLM explicitly needing to generate a corresponding free-form text*. As we show in Section 4.2, this enables the Bayes-optimal response to empirically outperform other decoding approaches and MBR in raw language space $\ell_{\text{Sample}}$.

## 2.2. Uncertainty Quantification Through Bayes Risk

Moreover, we show how to utilize the task-dependent structure $\mathcal{L}$ to quantify the uncertainty associated with a (latent) response. As discussed above, the mapping $g_T$ induces a distribution $p(\ell \mid x)$ over the latent space $\mathcal{L}$. We define the true uncertainty as the discrepancy between this predictive belief and the (unknown) true distribution $p^*(\ell \mid x)$ over latents. If $d_T$ is a proper metric, a natural notion of distance between these distributions is the 1-Wasserstein distance

$$W_1(p^*(\ell \mid x), p(\ell \mid x)) := \inf_{\pi \in \Pi(p^*, p)} \mathbb{E}_{(\ell, \ell') \sim \pi}[d(\ell, \ell')]$$

where $\Pi(p^*, p)$ denotes the set of all couplings (joint distributions) with marginals $p^*(\ell \mid x)$ and $p(\ell \mid x)$.

**Unambiguous Ground-Truth** A common assumption in supervised learning and uncertainty that most of the existing estimators in LLMs rely on is that there is no inherent ambiguity in the ground-truth (Tomov et al., 2025). This lack of inherent, so-called aleatoric uncertainty (Hüllermeier & Waegeman, 2021) can be understood as the reference LLM response for a given task to be concentrated at a single point in the latent space $\mathcal{L}$. Formally, the (unknown) reference distribution the LLM is ought to approximate well through $p(\ell \mid x)$ is described by a Dirac $p^*(\ell \mid x) = \delta(\ell = \ell^*)$ located at the true latent label $\ell^* \in \mathcal{L}$. In this case, the Wasserstein distance admits the closed form

$$W_1(p^*(\ell \mid x), p(\ell \mid x)) = \mathbb{E}_{\ell \sim p(\ell|x)}[d(\ell^*, \ell)] = R(\ell^*), \qquad (7)$$

which is precisely the Bayes Risk of the true action (response) $\ell^*$ under the model's predictive push-forward distribution and the task-dependent loss/dissimilarity $d$.

Since the Bayes action $\ell_{\text{Bayes}}$ minimizes the expected risk under $p(\ell \mid x)$, we obtain the inequality

$$R_{\text{Bayes}} = \min_{\tilde{\ell} \in \mathcal{L}} \mathbb{E}_{\ell \sim p(\ell|x)}\left[d(\tilde{\ell}, \ell)\right]$$
$$\leq \mathbb{E}_{\ell \sim p(\ell|x)}[d(\ell^*, \ell)] = R(\ell^*). \qquad (8)$$

Thus, the minimum Bayes risk is a *lower bound* on the true (epistemic) risk. Consequently, a large Bayes risk *provably implies* a large true risk[1]. This establishes $R_{\text{Bayes}}$ as a principled, model-internal, and task-dependent surrogate for uncertainty that is aware of the task's structure $\mathcal{L}$.

In Section 3.5, we introduce a framework to equip the latent structure $\mathcal{L}$ itself with a notion of (aleatoric) uncertainty to facilitate uncertainty estimation even if the ground-truth reference response is associated with inherent ambiguity. This way, we recover the assumptions of Equation (7) for tasks with aleatoric uncertainty.

## 3. Examples of Latent Spaces and Metrics

The framework of defining a task-dependent latent structure can be applied to many downstream applications of LLMs. It can be used in any scenario that admits a latent (possibly metric) space $(d_T, \mathcal{L})$ to embed the LLM's responses in. We instantiate this framework for several common LLM tasks (Table 1 & 9). For each task, we derive the corresponding closed-form Bayes-optimal action $\ell_{\text{Bayes}}$ and the associated uncertainty $R_{Bayes}$. We provide all proofs in Section D.

### 3.1. Single-Label Classification

A standard task in uncertainty quantification for LLMs (Kuhn et al., 2023; Nikitin et al., 2024; Duan et al., 2024) is classification over a set of $K$ classes. Let $[K] := \{1, \ldots, K\}$ denote the set of valid labels which can, for example, be estimated from the $M$ Monte-Carlo generations $s \mid x$. A suitable metric for this space is the *exact-match* (0–1) loss

$$d_T(\ell, \ell') := \mathbf{1}[\ell \neq \ell']. \qquad (9)$$

**Lemma 3.1.** *For any given class* $\ell \in [K]$, *let* $p_\ell = \Pr(g_T(s) = \ell)$ *denote the relative frequency among the MC samples. Under exact-match loss* $d_T(\hat{\ell}, \ell) = \mathbf{1}\{\hat{\ell} \neq \ell\}$, *a Bayes-optimal action is any mode of* $p$,

$$\ell_{Bayes} = \arg \max_{k \in [K]} \Pr(\ell = k) = \arg \max_{k \in [K]} p_k.$$

*The corresponding Minimum Bayes Risk is*

$$R_p(\ell_{Bayes}) := \mathbb{E}_{\ell \sim p}\left[\mathbf{1}\{\ell_{Bayes} \neq \ell\}\right] = 1 - \max_{k \in [K]} p_k.$$

Lemma 3.1 recovers the result that, under 0–1 loss, the Bayes-optimal prediction is the most probable class. The Bayes risk is exactly the residual probability mass outside the most likely class, $1 - \max_k p_k$, which quantifies the minimal achievable misclassification probability under $p$.

---

[1]This property even holds if $d$ is not a proper metric.

*Table 1.* Examples of latent spaces $\mathcal{L}$, task metrics $d$, and the resulting Bayes-optimal (or task-specified) decoding rules.

| Latent space $\mathcal{L}$ | Metric $d(\ell, \ell')$ | Bayes action $\ell_{\text{Bayes}}$ | Minimum Bayes Risk $R(l_{bayes})$ | Example task |
|---|---|---|---|---|
| Classes $[K]$ | $\mathbf{1}\{\ell \neq \ell'\}$ | $\arg\max_{i \in [K]} \Pr(\ell = i)$ | $1 - \max_{i \in [K]} \Pr(\ell = i)$ | Single-label QA / classification |
| Sets $\{0,1\}^K$ | $\sum_{i=1}^{K} \mathbf{1}[\ell_i \neq \ell'_i]$ | $\ell_i = \mathbf{1}[\Pr(\ell_i = 1 \mid x) \geq \frac{1}{2}]$ | $\sum_i \min\{\Pr(\ell_i = 1), \Pr(\ell_i = 0)\}$ | Multi-answer QA |
| Graphs $\mathcal{G} \cong \{0,1\}^E$ | $\sum_e \mathbf{1}[e \in E_\ell \veebar e \in E_{\ell'}]$ | $\ell_e = \mathbf{1}[\Pr(e \in E_\ell) \geq \frac{1}{2}]$ | $\sum_e \min\{\Pr(e \in E_\ell), \Pr(e \notin E_\ell)\}$ | Knowledge Graphs Extraction, Summarization |
| Semantic Embeddings $\{\ell \in \mathbb{R}^d : \|l\|_2 = 1\}$ | $1 - \langle \ell, \ell' \rangle$ | $\frac{\mathbb{E}_\ell[\ell]}{\|\mathbb{E}_\ell[\ell]\|}$ | $1 - \|\mathbb{E}_\ell[\ell]\|$ | Machine Translation |
| Probability simplex $\Delta^K$ | $\sum_k \ell_k \log \frac{\ell_k}{\ell_{k'}}$ | $\mathbb{E}_\ell[\ell]$ | $\mathbb{H}(\mathbb{E}_\ell[\ell]) - \mathbb{E}_\ell[\mathbb{H}(\ell)]$ | Single-label QA with predictive distributions |

## 3.2. Multi-Label Classification / Prediction Sets

One way to address a degree of ambiguity in a question that permits more than one correct answer is to model the latent LLM responses $\mathcal{L}$ as the power set over a finite number of classes $\mathcal{L} = 2^{[K]}$. Every response is mapped to an (potentially empty) set of labels in $[K]$. We illustrate this on an (ambiguous) question answering task. Consider:

> **Question:** Which ocean borders the USA?
> **Answers:** {Pacific, Atlantic}
> **Response 1:** The answer is Pacific Ocean
> **Response 2:** The answers are Pacific Ocean and Atlantic Ocean

Here, the $g_T$ maps the first answer to {Pacific} while the second response is mapped to {Pacific, Atlantic}. Each response $\ell$ can be identified by 1-hot encoded vectors $\{0,1\}^K$. A suitable distance metric is the Hamming loss

$$d_T(\ell, \ell') := \sum_{k=1}^{K} \mathbf{1}[\ell_k \neq \ell'_k]. \qquad (10)$$

**Lemma 3.2.** *Let $\ell \in \{0,1\}^K$ be distributed according to $p(\ell \mid x)$ and consider the Hamming loss $d_T(\ell, \ell') = \sum_k^K \mathbf{1}[\ell_k \neq \ell'_k]$. The Bayes-optimal prediction $\ell^{Bayes}$ is given component-wise by*

$$(\ell_{Bayes})_i = \mathbf{1}\{\Pr(\ell_i = 1) \geq \tfrac{1}{2}\}.$$

*The corresponding Minimum Bayes Risk is*

$$R(\ell_{Bayes}) = \sum_{k=1}^{K} \min\{\Pr(\ell_i = 1), \Pr(\ell_i = 0)\} \leq \tfrac{K}{2}.$$

Each potential response is included in the Bayes-optimal response if it appears in the majority of the sampled responses. The Bayes risk aggregates the uncertainty of whether each possible answer $\ell_i$ should (not) be included in the response. We also supplement the Bayes-optimal action and the associated risk under the $F_\beta$ loss in Section B.2.

## 3.3. Semantic Embedding Spaces

Another latent structure are *semantic embedding spaces*, for example embedding texts on the $d$-dimensional unit sphere: $\mathcal{L} = \{\ell \in \mathbb{R}^d : \|l\|_2 = 1\}$. A natural loss on $\mathcal{L}$ is the cosine distance, which simplifies to:

$$d_T(\ell, \ell') := 1 - \frac{\langle \ell, \ell' \rangle}{\|\ell\|_2 \|\ell'\|_2} = 1 - \langle \ell, \ell' \rangle. \qquad (11)$$

**Lemma 3.3.** *Let $\ell \in \mathbb{R}^d : \|l\|_2 = 1$ be distributed according to $p(\ell \mid x)$ with mean $\mu := \mathbb{E}_{\ell \sim p}[\ell]$. Consider the loss $d(\ell, \ell') = 1 - \langle \ell, \ell' \rangle$. If $\mu \neq 0$, the Bayes-optimal action is*

$$\ell_{Bayes} = \arg\min_{\ell \in \mathcal{L}} \mathbb{E}_{\ell \sim p}[1 - \langle \ell, \ell' \rangle] = \frac{\mu}{\|\mu\|},$$

*and the corresponding Minimum Bayes Risk is*

$$R(\ell_{Bayes}) = \mathbb{E}_{\ell \sim p}[1 - \langle \ell, \ell_{Bayes} \rangle] = 1 - \|\mu\|.$$

Lemma 3.3 shows that under cosine distance on the unit sphere, the Bayes-optimal semantic prediction is the *normalized mean embedding*. Moreover, the Bayes risk is fully characterized by the unnormalized mean embedding $\mu$ and how close it is to the sphere's surface: when $\|\mu\|$ is small, the mass under $p$ is dispersed over the sphere (high uncertainty), whereas a large $\|\mu\|$ indicates strong concentration (low uncertainty).

## 3.4. Knowledge Graphs

Text understanding, for example, in summarization tasks, often involves structuring its information (Jurafsky, 2014). Knowledge graphs $G = (V, E)$ keep track of different entities $\mathcal{V}$ within a text and their relations $\mathcal{E}$. For example, the sentence

> **Text**: The Eiffel Tower is in Paris, which is the capital of France.

introduces the entities {Paris, France, Eiffel Tower} and the relations "is capital of" and "is in". Consequently,

we can extract knowledge graphs from text paragraphs that define the relations over a set of entities $V$. While there are many potential mappings from token sequences $\mathcal{V}^\infty \mapsto \mathcal{G}$, here we discuss a simple approach that builds on the literature of knowledge graph extraction. In particular, we utilize KGGen (Mo et al., 2025) to obtain a set of annotated relations $E$ from a text excerpt, e.g. ("Paris", "is capital of", "France"). These relations imply the corresponding entities $V$ in the knowledge graph. Representing knowledge graphs $\ell$ with their relations $E_\ell$ is structurally isomorphic to the set-based latent structure of Section 3.2: The relations can be identified by 1-hot encoded vectors $\mathcal{L} \cong \{0,1\}^{|E|}$. Consequently, we use the structural Hamming distance to derive the Bayes-optimal action and its associated risk. The Bayes-optimal graph for a given LLM response distribution over relations (edges) $p(E)$ under the structural Hamming distance $d_T(\ell, \ell') = \sum_e \mathbf{1}[e \in E_\ell \veebar e \in E_{\ell'}]$ is given per-edge:

$$E_{\ell_{\text{Bayes}}} := \{e : \Pr[e \in E_\ell] \geq \frac{1}{2}\}$$
$$V_{\ell_{\text{Bayes}}} := \{v : v \in E_{\ell_{\text{Bayes}}}\}. \tag{12}$$

Analogous to Section 3.2, the corresponding Minimum Bayes Risk is

$$R(\ell_{\text{Bayes}}) = \sum_e \min\{\Pr[e \in E_\ell], \Pr[e \notin E_\ell]\} \tag{13}$$

Again, we include an edge (relation) if it is included in the majority of the sampled knowledge graphs and the risk aggregates the uncertainty of whether each individual edge should (not) be included in the Bayes-optimal graph.

### 3.5. Ambiguous Ground-Truth: Probability Simplex

Here, we describe how our framework can account for aleatoric uncertainty associated with the true reference LLM response. The core idea is to model aleatoric uncertainty in the latent structure by representing responses $\ell \in \mathcal{L}$ as probability distributions over individual outcomes in the downstream task. In multi-answer question answering, this can be achieved by prompting the LLM to verbalize its own belief (Tian et al., 2023) over possible answers:

**Question:** With what probability will it be sunny, cloudy, or rainy tomorrow?
**Response:** With $80\%$ probability it will be sunny tomorrow, with $15\%$ cloudy, and with $5\%$ rainy.

This response naturally corresponds to the probability vector $\ell = (0.8,\ 0.15,\ 0.05)^T$, where each coordinate represents the model's expressed belief in one of the three outcomes.

More generally, let $\mathcal{C}$ be the space of mutually exclusive outcomes for a given prompt, in the case of question answering, $\mathcal{C} = [K]$. We define $\mathcal{L}$ to be the space of all

so-called first-order probability measures over $\mathcal{C}$ similar to Sale et al. (2023). In classification, this space is isomorphic to the $(K-1)$-dimensional probability simplex $\mathbb{L} \cong \Delta^{K-1} := \{\ell \in [0,1]^K : \sum_k \ell_k = 1\}$. Each $\ell$ is a $K$-dimensional vector that assigns an outcome a probability. The more the probability mass is dispersed in $\ell$, the more aleatoric uncertainty is inherent to the LLM's response.

As the elements of the latent space are now distributions over answers, a suitable dissimilarity measure is the KL divergence $d_T(\ell, \ell') = \text{KL}[\ell \| \ell'] = \sum_k \ell_k \log \frac{\ell_k}{\ell'_k}$. Even though the KL divergence is not a metric, we can still obtain a closed-form solution for the Bayes-optimal action:

**Lemma 3.4.** *Let $\ell \in \Delta^{K-1}$ be distributed according to $p(\ell \mid x)$ and consider the KL divergence $d_T(\ell, \ell') = \sum_k \ell_k \log \frac{\ell_k}{\ell'_k}$. The Bayes-optimal prediction is the mean:*

$$\ell_k^{Bayes} = \mathbb{E}_\ell[\ell].$$

*The corresponding Minimum Bayes Risk is*

$$R(\ell_{Bayes}) = \mathbb{H}(\mathbb{E}_\ell[\ell]) - \mathbb{E}_\ell[\mathbb{H}(\ell)].$$

This latent structure applies to classification tasks with inherent ambiguity. Notably, the (unknown) reference latent response is still a point-mass on the true probability distribution and thus fulfills the assumptions of Equation (7) to enable uncertainty estimation.

## 4. Experiments

We experimentally study our framework of mapping LLM responses in a task-dependent latent space from two angles:[2] (i) Does the structure-aware Bayes-optimal response $\ell_{\text{Bayes}}$ outperform other decoding schemes over different evaluation metrics? (ii) Does uncertainty as the structure-aware Bayesian risk outperform existing uncertainty estimators in terms of indicating the quality of the generation?

### 4.1. Experimental Setup

**Tasks and Latent Structures.** We test our framework on multiple tasks $T$ and latent structures $\mathcal{L}$ that are equipped with corresponding measures of dissimilarity $d_T$. We mainly use algorithmic post-processing and auxiliary language models to implement $g_T$ and map language outputs to their respective latent structures (see Section B.1). The mappings $g_T$ often represent relatively easy tasks that we find can be reliably addressed with external language models if needed.

(i) **Classification**: We consider the single-answer question answering dataset TriviaQA (Joshi et al., 2017) and map

---

[2]We provide code for our implementation at https://github.com/dfuchsgruber/task_awareness_in_llms.

*Table 2.* Generation quality of structure-aware MBR (ours) versus other decoding strategies, including the sampled LLM response with minimal Bayes risk $\ell_{\text{Sample}}$ (**best**). In latent space similarity and task-specific metrics, ours matches or outperforms other decoding.

| Dataset (Latent $\mathcal{L}$) | | MAQA ($\{0,1\}^K$) Hamming ↓ | F1 ↑ | TriviaQA ($[K]$) Exact Match ↓ | WMT19 ($\mathbb{R}^d$) COMET ↑ | Cosine ↓ | Summarization ($\mathcal{G}$) Hamming ↓ | F1 ↑ | Align Score ↑ | MAQA ($\Delta^{K-1}$) KL ↓ |
|---|---|---|---|---|---|---|---|---|---|---|
| **Gemma-3-4B** | Beam | 0.892 | 0.548 | **0.627** | **0.848** | 0.131 | 2.176 | 0.192 | 0.831 | 10.330 |
| | Self-Consistency | 0.892 | 0.558 | 0.629 | **0.848** | 0.131 | 2.212 | 0.183 | 0.852 | 10.192 |
| | Contr. | 1.341 | 0.518 | 0.675 | 0.791 | 0.149 | 2.323 | 0.221 | 0.809 | 10.826 |
| | AR | 0.883 | 0.565 | 0.638 | 0.846 | 0.132 | 2.207 | 0.187 | 0.852 | 10.507 |
| | Greedy | 0.927 | 0.556 | 0.632 | 0.847 | 0.131 | 2.097 | **0.229** | 0.828 | 10.027 |
| | $\ell_{\text{MAP}}$ | 0.856 | 0.559 | | **0.848** | 0.131 | 2.069 | 0.188 | 0.832 | 10.545 |
| | $\ell_{\text{Sample}}$ | 0.838 | 0.570 | 0.632 | **0.848** | 0.130 | 1.930 | 0.211 | 0.849 | 10.514 |
| | Bayes (ours) | **0.729** | **0.576** | | **0.848** | **0.124** | **1.603** | 0.224 | **0.876** | **7.940** |
| **Gemma-3-12B** | Beam | 0.674 | **0.684** | 0.417 | **0.863** | 0.115 | 2.051 | 0.203 | 0.843 | 7.792 |
| | Self-Consistency | 0.706 | 0.678 | 0.418 | **0.863** | 0.114 | 2.071 | 0.200 | 0.860 | 7.980 |
| | Contr. | 0.762 | 0.661 | 0.423 | **0.863** | 0.115 | 1.945 | **0.235** | 0.860 | 7.689 |
| | AR | 0.717 | 0.673 | 0.424 | 0.862 | 0.115 | 2.085 | 0.210 | 0.861 | 8.104 |
| | Greedy | 0.714 | 0.673 | 0.418 | **0.863** | 0.115 | 2.020 | 0.214 | 0.845 | 7.859 |
| | $\ell_{\text{MAP}}$ | 0.692 | 0.677 | | **0.863** | 0.114 | 1.955 | 0.186 | 0.844 | 8.051 |
| | $\ell_{\text{Sample}}$ | 0.674 | 0.680 | **0.416** | 0.863 | 0.114 | 1.846 | 0.203 | 0.869 | 8.420 |
| | Bayes (ours) | **0.601** | **0.684** | | 0.863 | **0.109** | **1.597** | 0.231 | **0.889** | **5.632** |
| **Qwen3-4B** | Beam | 0.953 | 0.509 | **0.632** | **0.843** | 0.136 | 2.238 | 0.208 | 0.849 | 12.786 |
| | Self-Consistency | 0.975 | 0.499 | 0.634 | **0.843** | 0.137 | 2.258 | 0.201 | 0.842 | 12.902 |
| | Contr. | 1.212 | 0.479 | 0.645 | **0.843** | 0.137 | 2.226 | 0.228 | 0.833 | 12.373 |
| | AR | 1.022 | 0.505 | 0.648 | 0.842 | 0.136 | 2.253 | 0.205 | 0.838 | 13.297 |
| | Greedy | 1.045 | 0.503 | 0.636 | **0.843** | 0.137 | 2.310 | **0.239** | 0.825 | 12.470 |
| | $\ell_{\text{MAP}}$ | 0.913 | 0.504 | | 0.842 | 0.137 | 2.206 | 0.180 | 0.846 | 12.885 |
| | $\ell_{\text{Sample}}$ | 0.917 | 0.508 | 0.635 | 0.843 | 0.136 | 2.074 | 0.214 | 0.851 | 13.145 |
| | Bayes (ours) | **0.858** | **0.511** | | 0.843 | **0.129** | **1.851** | 0.222 | **0.869** | **10.415** |
| **Qwen3-30B-A3B** | Beam | 0.581 | 0.695 | 0.415 | 0.863 | 0.113 | 2.422 | 0.183 | 0.841 | 6.759 |
| | Self-Consistency | 0.588 | 0.686 | 0.419 | 0.863 | 0.114 | 2.430 | 0.193 | 0.807 | 7.055 |
| | Contr. | 0.719 | 0.662 | 0.428 | 0.859 | 0.114 | 2.311 | **0.216** | 0.836 | 7.299 |
| | AR | 0.603 | 0.681 | 0.424 | 0.863 | 0.114 | 2.463 | 0.183 | 0.815 | 6.873 |
| | Greedy | 0.623 | 0.690 | 0.416 | **0.864** | 0.113 | 2.403 | 0.193 | 0.804 | 6.920 |
| | $\ell_{\text{MAP}}$ | 0.563 | 0.694 | | 0.863 | 0.114 | 2.301 | 0.187 | 0.837 | 6.884 |
| | $\ell_{\text{Sample}}$ | 0.551 | 0.695 | **0.414** | 0.863 | 0.113 | 2.249 | 0.192 | 0.840 | 7.283 |
| | Bayes (ours) | **0.525** | **0.698** | | 0.863 | **0.109** | **2.008** | 0.197 | **0.863** | **5.032** |

responses to one of the finite classes $\mathcal{L} = [K]$. We evaluate the Bayes-optimal response and the associated Bayes risk under the exact-match loss. (ii) **Sets**: To demonstrate set-structured responses, we use multi-answer question answering (MAQA (Yang et al., 2025)), where the latent $\ell \in \{0, 1\}^K$ represent sets of valid answers. We study both the Hamming distance and the F1 score as dissimilarity measures. (iii) **Undirected Graphs**: We represent summaries as undirected knowledge graphs $(V, E)$ extracted from generated text, following standard procedures from the speech processing literature (Jurafsky, 2014). We evaluate abstractive summarization on a CNN/DailyMail-based benchmark dataset (See et al., 2017; Zhang et al., 2023) using the structural Hamming distance to compute risk. The generations and uncertainty are evaluated with respect to Hamming distance, F1 score, and the AlignScore (Zha et al., 2023). The latter requires reassembling raw texts as described in Section B.1. (iv) **Semantic Space** $\mathcal{S}$: Here, we consider machine translation on WMT19 (FI-EN) (Barrault et al., 2019) by embedding the translations of the LLM into a semantic latent space. We use the cosine similarity to compute Bayesian risk, but also evaluate the quality of the generations and uncertainty using the COMET score(Rei et al., 2020). As this requires raw texts, we compute the COMET score for the closest sampled LLM response in

terms of $d_T$. (v) **Probability Simplex**: Lastly, to demonstrate absorbing aleatoric uncertainty into the task structure $\mathcal{L}$, we study multi-answer question answering with ground-truth probability distributions over the answers. We use the recently proposed annotations on MAQA (Tomov et al., 2025). The KL-divergence serves as latent distance $d_T$.

**Models.** We evaluate instruction-tuned versions of Gemma 3 4B, Gemma 3 12B (Gemma Team, 2025), Qwen 3 4B, and Qwen 3 30B A3E (Qwen Team, 2025). All models are used with default decoding parameters (temperature, top-$p$, and top-$k$), reflecting standard deployment settings. Prompts can be found in Section E. We draw $M = 20$ samples as surrogates from the push-forward distribution $p(\ell \mid x)$ as in Equation (5).

### 4.2. Bayes-Optimal Decoding in Latent Space

First, we compare the structure-aware Bayes optimal responses $\ell_{\text{Bayes}}$ to other decoding paradigms under different task-dependent metrics. We compare our approach to beam search, self-consistency decoding (Wang et al., 2023), contrastive search (Contr.) (Su et al., 2022), standard autoregressive sampling (AR), greedy decoding, and the MAP of the latent belief $\ell_{\text{MAP}}$ as well as the sampled response with minimal Bayes risk $\ell_{\text{sample}}$. We study both the dis-

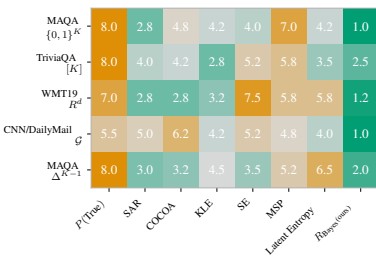
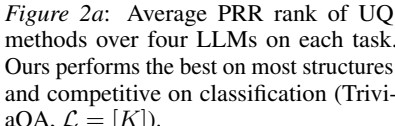

*Figure 2a*: Average PRR rank of UQ methods over four LLMs on each task. Ours performs the best on most structures and competitive on classification (TriviaQA, $\mathcal{L} = [K]$).

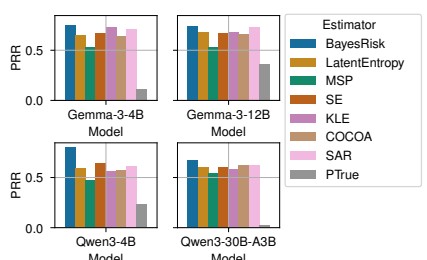
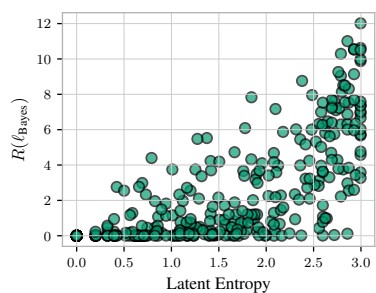

*Figure 2b*: PRR for each UQ method on MAQA ($\mathcal{L} = \{0, 1\}^K$). Our Bayes risk correlates the most with the generation quality (F1 score).

*Figure 2c*: Entropy in the latent distribution $p(\ell \mid x)$ versus the Minimum Bayes Risk on MAQA ($\Delta^{K-1}$). The latter takes low values even for high entropy distributions as it is distance-aware through $d_T$.

similarity between the latent response and the reference answer in the latent space (Hamming, Cosine, F1, Exact Match, KL) as well as free-form text metrics like COMET score or AlignScore. Table 2 shows that the Bayes-optimal response outperforms other decoding strategies over various tasks. For single-answer classification ($\mathcal{L} = [K]$) on TriviaQA, the Bayes-optimal action, $\ell_{\text{Sample}}$, and $\ell_{\text{MAP}}$ coincide. On this simple structure, the most likely latent class $\ell_{\text{Bayes}} \in [K]$ typically aligns with the high likelihood raw language response of beam search decoding. Overall, the Bayes-optimal responses are of higher quality compared to the self-consistency decoding baseline (Wang et al., 2023) and responding with the majority vote among the sampled latent responses ($\ell_{\text{MAP}}$). We also find it to outperform MBR in the space of raw language $\ell_{\text{Sample}}$ as defined in Equation (6). This highlights the power of obtaining the optimal LLM response in closed form *in the latent space* as $\ell_{\text{Bayes}}$ outperforms the lowest risk MC sample among the responses.

We illustrate this on an example for $\mathcal{L}$ containing sets of possible answers as described in Section 3.2. The Bayes-optimal response for the Hamming distance metric includes an element $y$ only if the majority of MC samples include $y$ as well. If the LLM is uncertain, these samples are likely disjunct sets of different elements $y$. For example, asking for "Which oceans border the USA?" may result in the latent responses {Pacific}, {Atlantic}, {Indian}. As none of the outcomes is included in the majority of the MC samples, the Bayes-optimal prediction will be the empty set even if every individual MC sample $\ell^{(i)}$ is itself non-empty. Here, the latent structure enables LLM to abstain from a prediction if the associated uncertainty (variation in the push-forward latent distribution) is too large.

**Bayes-Optimal Decoding Improves Generations under Uncertainty** We also study what drives the performance of the Bayes-optimal response $\ell_{\text{Bayes}}$. Figure 3 shows that we observe the most performance gains over other decoding strategies in Hamming distance to the reference response $\Delta = d(\ell^*, \ell_{\text{Beam}}) - d(\ell^*, \ell_{\text{Bayes}})$ when the latent

push-forward distribution $p(\ell \mid x)$ is high. This pattern is consistent across models and tasks (Section A.3). At low entropy, the MC samples are similar and coincide with beam search and the Bayes-optimal response. As the entropy increases, $\ell_{\text{Bayes}}$ can account for this variability by using a centroid response, while beam search and MBR output one of the many low-probability answers. This shows that the Bayes-optimal action outperforms other decoding methods, particularly when the LLM is uncertain about its response.

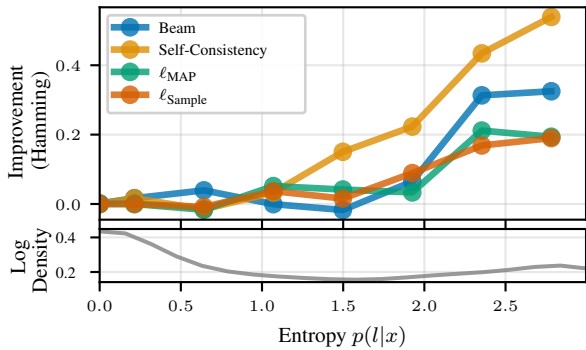

*Figure 3.* Improvement (Hamming distance) of our Bayes-optimal prediction over each other decoding baseline for set-based multi-answer QA (MAQA) vs. the entropy of the push-forward latent distribution $p(\ell \mid x)$. Under high variability, we synthesize answers that are substantially different from other outputs.

### 4.3. Uncertainty Quantification via Bayes Risk

**Baselines.** We consider the common uncertainty estimators: Maximum sentence probability (MSP), semantic entropy (SE) (Kuhn et al., 2023), kernel language entropy (KLE) (Nikitin et al., 2024), shifting attention to relevance (SAR) (Duan et al., 2024), CoCoA (Vashurin et al., 2025b) and $p(\text{True})$ (Kadavath et al., 2022). We also compare to using the entropy of the latent distribution $H(p(\ell|x))$.

**Evaluation metrics.** We evaluate uncertainty estimates using Prediction Rejection Ratio (PRR)(Malinin & Gales, 2021; Vashurin et al., 2025a) and concordance AUC (AUC) (Therneau & Atkinson, 2024). Both measure the align-

ment between an uncertainty estimate and the actual task-dependent quality metric (e.g. Hamming distance, Exact-Match / Accuracy, Align Score, COMET, KL divergence). Higher values indicate better alignment between estimated uncertainty and task-level performance.

**Results.** Figure 2a ranks each uncertainty estimator per model and task in terms of the PRR and averages over the four LLMs. On 4 out of 5 structures, the risk-based uncertainty of our framework predicts the response quality with respect to the task-dependent performance metric the best. On classification ($\mathcal{L} = [K]$), the task existing estimators are explicitly designed for, our framework performs competitively. Many of the baseline approaches are based on a notion of semantic variability among the MC LLM responses. Only our structure-aware Bayesian risk explicitly accounts for (i) A task-dependent *structural variability*, and (ii) the *spread* of the distribution in terms of the structural task-aware distance $d_T$. We explicitly visualize the PRR for the set structure $\mathcal{L} = \{0,1\}^K$ in Figure 2b and supply the corresponding AUC and PRR values for each model in Section A.2. We observe substantial improvements of our structure-aware uncertainty estimate over the distance-agnostic variability-based methods like Semantic Entropy (SE). Figure 2c shows how Bayesian risk goes beyond simply measuring variability in the response's semantics and factors in how different they are in terms of the task-dependent structural dissimilarity $d_T$.

## 5. Related Work

**MBR decoding in LLMs** MBR decoding has primarily been studied in machine translation, where expected risk is estimated using a learned similarity such as BERTScore or BLEURT to select from candidate outputs (Eikema & Aziz, 2022; Müller & Sennrich, 2021). Bertsch et al. (2023) show that different decoding strategies, including self-consistency (Wang et al., 2023), can be interpreted as implicitly minimizing Bayes risk as well. They observe that MBR is improved when using the same distance $d$ for decoding and in evaluation. Recent work extends MBR beyond translation: Wu et al. (2024) use an LLM judge as a utility function to select higher-quality responses, while Lukasik et al. (2024) generalize MBR to regression settings, showing that Bayesian-optimal actions under squared- and absolute-error losses outperform beam search. Concurrently with our work, Eikema et al. (2025) modify existing similarity metrics based on task-specific output structures. However, they do not map responses to a structured latent domain to synthesize Bayes-optimal outputs and instead rely on costly neural metrics in the space of raw language.

**UQ in LLMs** A wide range of uncertainty quantification (UQ) methods for LLMs has been proposed (Vashurin et al., 2025a; Liu et al., 2025). Existing approaches broadly fall into three categories: *information-based* methods that analyze token-level predictive distributions (e.g., MSP); *sample-diversity* methods that generate multiple outputs and assess their semantic variability, optionally incorporating probability estimates (Kuhn et al., 2023; Duan et al., 2024; Nikitin et al., 2024); and *verbalized* or reflexive methods that prompt models to explicitly express uncertainty (Kadavath et al., 2022). With the exception of verbalized approaches which often perform worst empirically (Vashurin et al., 2025a) most methods consider responses only as raw language or in a semantic space. Our risk-based estimator fully exploits the task's structure by leveraging the distance $d_T$ tailored to the specific use-case.

Previous work also studied uncertainty in the context of Bayesian risk: Vashurin et al. (2025b) use MBR to enhance sample-diversity methods by incorporating confidence measures such as maximum sentence probability. Similar to our work, Smith et al. (2025) propose to interpret minimum Bayes risk as a principled uncertainty measure to which we draw connections in Section C.1, while Wang & Holmes (2024) proposes risk-based uncertainty as well. ,We show that this interpretation of Bayesian risk as uncertainty is particularly effective when directly incorporating the task's latent structure through the distance function $d_T$. Lastly, in Section C.2 we set our UQ definitions into perspective with an information theoretic Bayesian decomposition of epistemic and aleatoric uncertainty (Depeweg et al., 2018; Wimmer et al., 2023).

## 6. Discussion

**Limitations** Our framework generalizes to arbitrary latent structures but its efficiency hinges on an easy-to-compute Bayes optimal action. Another bottleneck may be the mapping from language to the latents $g_T$, even though such post-processing is typically done anyway when systematically using LLM responses. For some tasks, we also may require free-form text responses after all, and constructing an inverse mapping $g_T^{-1}$ from latent actions back to natural language may not be feasible to compute. Inverse mappings $g_T^{-1}$ could also be constructed algorithmically, for example by relying on auxiliary LLMs. We want to stress that our framework is most useful for tasks in which the concrete latent response is more useful than the linguistic form.

We show that the generations of our framework improve over existing decoding methods when the LLM outputs diverse responses under uncertainty. This mild assumption is made by most output-informed or sampling-based uncertainty estimators but may not apply to every task to the same extent. Another assumption is that the LLM's responses can be mapped to the latent domain at all. We only empirically validate this assumption for strong instruction-tuned

LLMs. Our method requires access to the predictive distribution through MC samples which can be computationally demanding in certain applications, similar to existing MBR frameworks and uncertainty estimators. This can be mitigated by reducing the sample size which we find to still yield good results in Section A.1.

**Towards Task-Awareness** Our results suggest that LLM generation and uncertainty quantification can be improved by making the task's structure explicit through an appropriate latent representation. When responses are interpreted in a task-aware space, Minimum Bayes Risk decoding enables the synthesis of Bayes-optimal actions that improve output quality beyond generations obtained in the space of natural language. Moreover, the resulting Bayes risk provides a principled, task-aligned uncertainty measure that consistently tracks response quality better than existing estimators. Hence, we believe that our framework offers a highly general paradigm that facilitates the reliable integration of LLMs across a wide range of different downstream tasks. The perspective of interpreting an LLM's outputs in a task-aware way extends beyond generation quality and uncertainty estimation and can serve as a conceptual framework for addressing other open problems in LLM research in a principled and application-oriented way.

## Impact Statement

In this work, we examine how generation performance and uncertainty quantification in Large Language Models can be improved. While any research may be misused, our primary goal is to improve the reliability of these models to support their safe deployment in critical domains. We believe the benefits will outweigh the potential risks.

## Acknowledgements

We want to give special thank to Soroush H. Zargarbashi for giving helpful suggestions on an early draft of the manuscript. We also want to thank Franz Rieger, Niklas Kemper, David Lüdke, and Filippo Guerranti for reviewing the paper. The research presented has been performed in the frame of the RADELN project funded by TUM Georg Nemetschek Institute Artificial Intelligence for the Built World (GNI). It is further supported by the Bavarian Ministry of Economic Affairs, Regional Development and Energy with funds from the Hightech Agenda Bayern.

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

# A. Additional Experiments

## A.1. Computational Ablations

As expected, the quality of both the Bayes-optimal decoding action and the Bayes risk as an uncertainty estimate depends on the number of samples used to approximate the model-induced distribution. In Figure 4, we analyze this dependence for decoding performance on set-structured outputs under Hamming loss. Performance consistently improves with increasing sample size and, notably, even a small number of samples suffices to outperform beam search, indicating that substantial gains can be achieved with relatively low additional computational cost.

For uncertainty quantification, the dependence of the Bayes-optimal output $\ell_{\text{Bayes}}$ on the number of samples introduces an interaction effect that leads to stable and consistent behavior as the sample size grows (Figure 5). When the output is fixed, for example, to beam search, we again observe a trend similar to that of decoding performance, with uncertainty estimates improving steadily as the number of samples increases (Figure 6).

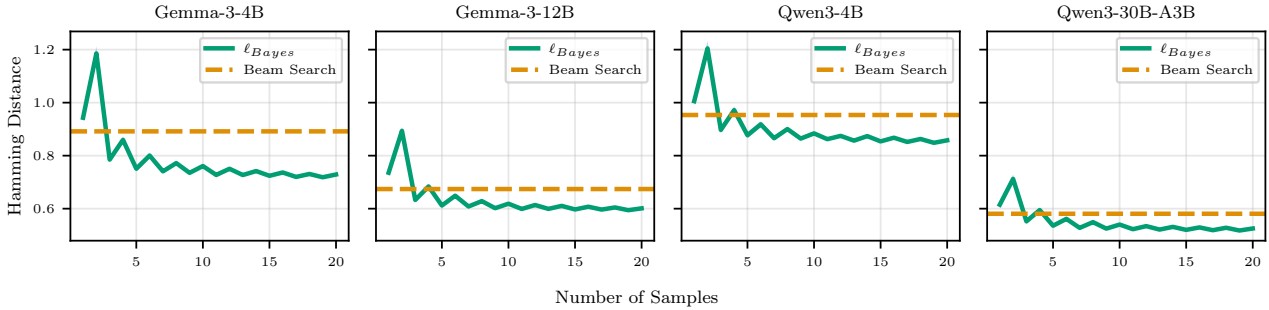

*Figure 4.* Performance in Hamming Distance of $\ell_{Bayes}$ estimator on Multi-Answer QA over an increasing numbers of MC samples.

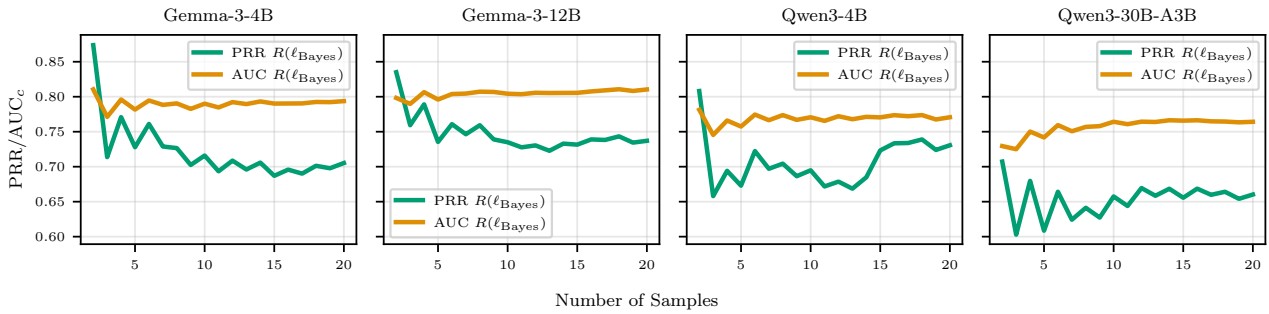

*Figure 5.* UQ performance of $R(\ell_{Bayes})$ on Multi-Answer QA over an increasing numbers of MC samples.

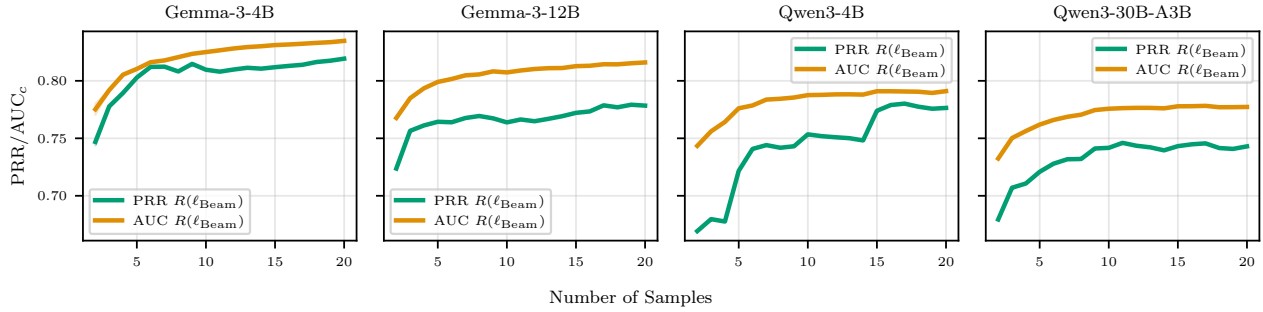

*Figure 6.* UQ performance of $R(\ell_{Beam})$ on Multi-Answer QA over an increasing numbers of MC samples.

## A.2. Uncertainty Estimation

We supply the performance of different uncertainty estimators, including our Bayesian risk $R(\ell_{\text{Bayes}})$ in terms of both PRR and AUC in Tables 3 to 6. To that end, we measure the alignment between each respective proxy for uncertainty and the

*Table 3.* Uncertainty quantification PRR ↑ for different estimators on all tasks (best and runner-up). We measure how well the uncertainty aligns with metrics computed from the Bayes optimal response.

| Dataset | | MAQA $\{0,1\}^K$ | | TriviaQA $[K]$ | WMT19 $R^d$ | | CNN/DailyMail $\mathcal{G}$ | | | MAQA $\Delta^{K-1}$ |
| --- | --- | --- | --- | --- | --- | --- | --- | --- | --- | --- |
| | | Hamming | F1 | Exact Match | COMET | Cosine | Hamming | F1 | Align Score | KL Divergence |
| Gemma-3-4B | $P(\text{True})$ | 0.148 | 0.134 | 0.343 | 0.082 | −0.036 | −0.182 | −0.207 | −0.031 | −0.147 |
| | SAR | 0.617 | 0.743 | 0.569 | **0.336** | *0.266* | 0.191 | **0.388** | 0.109 | *0.732* |
| | COCOA | 0.581 | 0.688 | 0.573 | 0.286 | 0.210 | *0.222* | 0.222 | 0.153 | 0.694 |
| | KLE | 0.675 | *0.749* | *0.697* | 0.319 | 0.161 | 0.099 | 0.044 | 0.041 | 0.697 |
| | SE | 0.776 | 0.719 | 0.666 | 0.146 | 0.079 | 0.174 | 0.199 | 0.050 | 0.703 |
| | MSP | 0.450 | 0.571 | 0.512 | 0.161 | 0.120 | 0.110 | 0.106 | 0.175 | 0.645 |
| | Latent Entropy | *0.804* | 0.705 | **0.713** | 0.232 | −0.006 | 0.137 | 0.170 | *0.240* | 0.682 |
| | $R_{\text{Bayes}(\text{ours})}$ | **0.819** | **0.811** | 0.674 | *0.327* | **0.282** | **0.677** | *0.355* | **0.289** | **0.754** |
| Gemma-3-12B | $P(\text{True})$ | 0.420 | 0.383 | 0.250 | 0.151 | −0.031 | −0.186 | −0.031 | −0.096 | 0.063 |
| | SAR | 0.697 | *0.726* | **0.589** | *0.270* | *0.199* | 0.098 | *0.207* | *0.130* | *0.652* |
| | COCOA | 0.655 | 0.655 | 0.568 | 0.263 | 0.149 | 0.096 | 0.181 | 0.052 | 0.632 |
| | KLE | 0.670 | 0.686 | *0.582* | 0.236 | 0.097 | 0.079 | 0.016 | −0.012 | 0.614 |
| | SE | 0.721 | 0.672 | 0.566 | 0.081 | 0.035 | *0.199* | 0.140 | 0.023 | 0.620 |
| | MSP | 0.504 | 0.512 | 0.539 | 0.180 | 0.105 | 0.013 | 0.184 | −0.004 | 0.590 |
| | Latent Entropy | *0.767* | 0.678 | 0.577 | 0.220 | −0.066 | 0.151 | 0.177 | 0.119 | 0.613 |
| | $R_{\text{Bayes}(\text{ours})}$ | **0.778** | **0.726** | 0.576 | **0.316** | **0.263** | **0.581** | 0.258 | **0.398** | **0.711** |
| Qwen3-4B | $P(\text{True})$ | 0.378 | 0.237 | 0.508 | 0.221 | *0.209* | 0.050 | 0.014 | 0.128 | 0.124 |
| | SAR | 0.464 | 0.590 | 0.519 | 0.235 | 0.177 | *0.299* | *0.200* | 0.140 | *0.516* |
| | COCOA | 0.398 | 0.569 | 0.516 | *0.276* | 0.199 | 0.231 | **0.252** | *0.153* | 0.498 |
| | KLE | 0.286 | 0.560 | 0.573 | 0.249 | 0.125 | 0.191 | 0.150 | 0.006 | 0.444 |
| | SE | 0.391 | *0.644* | 0.547 | 0.143 | 0.063 | 0.131 | 0.046 | 0.016 | 0.465 |
| | MSP | 0.328 | 0.483 | 0.510 | 0.200 | 0.184 | 0.070 | 0.187 | 0.084 | 0.498 |
| | Latent Entropy | *0.585* | 0.597 | *0.612* | 0.147 | −0.046 | 0.154 | 0.021 | 0.019 | 0.492 |
| | $R_{\text{Bayes}(\text{ours})}$ | **0.776** | **0.788** | **0.631** | **0.309** | **0.272** | **0.487** | 0.152 | **0.272** | **0.569** |
| Qwen3-30B-A3B | $P(\text{True})$ | 0.024 | 0.042 | 0.261 | 0.026 | −0.014 | −0.070 | 0.013 | 0.001 | 0.180 |
| | SAR | 0.664 | *0.644* | 0.503 | 0.247 | 0.172 | *0.343* | **0.471** | 0.012 | *0.672* |
| | COCOA | 0.667 | 0.632 | **0.545** | *0.285* | *0.185* | 0.299 | *0.404* | 0.032 | 0.670 |
| | KLE | 0.610 | 0.613 | 0.506 | 0.278 | 0.117 | 0.217 | 0.322 | −0.115 | 0.627 |
| | SE | 0.648 | 0.629 | 0.500 | 0.063 | 0.048 | 0.225 | 0.344 | −0.078 | 0.652 |
| | MSP | 0.549 | 0.557 | *0.521* | 0.190 | 0.151 | 0.271 | 0.300 | *0.119* | 0.615 |
| | Latent Entropy | *0.701* | 0.617 | 0.483 | 0.157 | −0.062 | 0.115 | 0.174 | −0.128 | 0.630 |
| | $R_{\text{Bayes}(\text{ours})}$ | **0.743** | **0.691** | 0.513 | **0.312** | **0.252** | **0.667** | 0.260 | **0.251** | **0.707** |

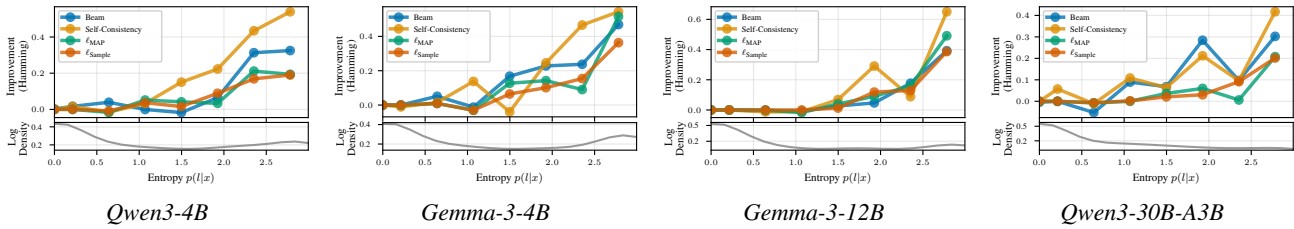

| *Qwen3-4B* | *Gemma-3-4B* | *Gemma-3-12B* | *Qwen3-30B-A3B* |

*Figure 7.* Improvement of $\ell_{bayes}$ across increasing latent entropy on multi-answer QA.

task-dependent performance metric. We also study if our uncertainty aligns with the performance of (i) our Bayes-optimal response $\ell_{\text{Bayes}}$ and (ii) the beam search response. In Tables 3 and 4, we find that both in terms of AUC and PRR, our risk-based uncertainty outperforms in most metrics and tasks or is at least competitive in terms predicting the performance of the Bayes-optimal action $\ell_{\text{Bayes}}$. Similarly, even when investigating the performance of the beam search response, Bayesian risk gives a strong performing estimate of the associated uncertainty as depicted in Tables 3 and 6. These findings highlight the merits of our risk-based estimate of uncertainty as it correlates well with the performance of the LLM's output.

### A.3. Decoding

In Figures 7 to 9, we illustrate the impact of latent entropy on decoding performance relative to $\ell_{\text{Bayes}}$ across models for Multi-Answer QA, Machine Translation, and Multi-Answer QA with simplex-structured outputs. The observed trends are consistent with the findings in Section 4.2 and, importantly, generalize across model families and output structures.

*Table 4.* Uncertainty quantification AUC ↑ for different estimators on all tasks (best and runner-up). We measure how well the uncertainty aligns with metrics computed from the Bayes optimal response.

| Dataset | | MAQA $\{0,1\}^K$ | | TriviaQA $[K]$ | WMT19 $R^d$ | | CNN/DailyMail $\mathcal{G}$ | | | MAQA $\Delta^{K-1}$ |
|---|---|---|---|---|---|---|---|---|---|---|
| | | Hamming | F1 | Exact Match | COMET | Cosine | Hamming | F1 | Align Score | KL Divergence |
| Gemma-3-4B | $P$(True) | 0.557 | 0.548 | 0.609 | 0.537 | 0.484 | 0.468 | 0.504 | 0.459 | 0.465 |
| | SAR | 0.734 | 0.784 | 0.741 | 0.608 | 0.554 | 0.449 | 0.561 | 0.514 | 0.709 |
| | COCOA | 0.712 | 0.751 | 0.779 | 0.590 | 0.536 | 0.449 | 0.536 | 0.513 | 0.698 |
| | KLE | 0.762 | 0.791 | 0.772 | 0.609 | 0.541 | 0.494 | 0.505 | 0.522 | 0.708 |
| | SE | 0.777 | 0.772 | 0.775 | 0.543 | 0.524 | 0.468 | 0.496 | 0.521 | 0.724 |
| | MSP | 0.676 | 0.710 | 0.764 | 0.565 | 0.521 | 0.455 | 0.521 | 0.508 | 0.685 |
| | Latent Entropy | 0.788 | 0.769 | 0.784 | 0.603 | 0.494 | 0.490 | 0.546 | 0.544 | 0.721 |
| | $R_{\text{Bayes}}$(ours) | 0.794 | 0.803 | 0.780 | 0.608 | 0.565 | 0.600 | 0.584 | 0.580 | 0.744 |
| Gemma-3-12B | $P$(True) | 0.670 | 0.654 | 0.644 | 0.557 | 0.494 | 0.497 | 0.420 | 0.487 | 0.539 |
| | SAR | 0.775 | 0.789 | 0.773 | 0.586 | 0.544 | 0.461 | 0.573 | 0.461 | 0.704 |
| | COCOA | 0.756 | 0.761 | 0.812 | 0.585 | 0.526 | 0.466 | 0.568 | 0.453 | 0.689 |
| | KLE | 0.771 | 0.772 | 0.767 | 0.588 | 0.524 | 0.462 | 0.523 | 0.469 | 0.701 |
| | SE | 0.768 | 0.762 | 0.761 | 0.519 | 0.508 | 0.519 | 0.563 | 0.486 | 0.708 |
| | MSP | 0.716 | 0.719 | 0.802 | 0.571 | 0.519 | 0.449 | 0.548 | 0.453 | 0.668 |
| | Latent Entropy | 0.803 | 0.782 | 0.765 | 0.600 | 0.481 | 0.477 | 0.540 | 0.530 | 0.704 |
| | $R_{\text{Bayes}}$(ours) | 0.810 | 0.800 | 0.764 | 0.595 | 0.562 | 0.580 | 0.571 | 0.605 | 0.753 |
| Qwen3-4B | $P$(True) | 0.590 | 0.568 | 0.640 | 0.525 | 0.550 | 0.495 | 0.477 | 0.522 | 0.555 |
| | SAR | 0.721 | 0.736 | 0.733 | 0.582 | 0.531 | 0.462 | 0.549 | 0.501 | 0.639 |
| | COCOA | 0.717 | 0.726 | 0.774 | 0.594 | 0.537 | 0.461 | 0.572 | 0.495 | 0.639 |
| | KLE | 0.701 | 0.725 | 0.738 | 0.596 | 0.523 | 0.428 | 0.537 | 0.479 | 0.620 |
| | SE | 0.712 | 0.746 | 0.714 | 0.535 | 0.512 | 0.451 | 0.523 | 0.508 | 0.624 |
| | MSP | 0.692 | 0.694 | 0.769 | 0.576 | 0.536 | 0.464 | 0.563 | 0.500 | 0.639 |
| | Latent Entropy | 0.751 | 0.732 | 0.736 | 0.584 | 0.487 | 0.483 | 0.544 | 0.511 | 0.614 |
| | $R_{\text{Bayes}}$(ours) | 0.771 | 0.809 | 0.737 | 0.605 | 0.558 | 0.525 | 0.552 | 0.594 | 0.639 |
| Qwen3-30B-A3B | $P$(True) | 0.499 | 0.505 | 0.625 | 0.518 | 0.503 | 0.522 | 0.533 | 0.442 | 0.529 |
| | SAR | 0.752 | 0.743 | 0.739 | 0.584 | 0.530 | 0.484 | 0.599 | 0.479 | 0.689 |
| | COCOA | 0.748 | 0.738 | 0.810 | 0.598 | 0.541 | 0.498 | 0.608 | 0.466 | 0.705 |
| | KLE | 0.733 | 0.719 | 0.736 | 0.600 | 0.521 | 0.496 | 0.563 | 0.507 | 0.688 |
| | SE | 0.742 | 0.724 | 0.728 | 0.517 | 0.508 | 0.414 | 0.525 | 0.466 | 0.700 |
| | MSP | 0.718 | 0.711 | 0.803 | 0.579 | 0.535 | 0.486 | 0.594 | 0.473 | 0.681 |
| | Latent Entropy | 0.764 | 0.734 | 0.735 | 0.589 | 0.485 | 0.482 | 0.597 | 0.483 | 0.698 |
| | $R_{\text{Bayes}}$(ours) | 0.764 | 0.752 | 0.734 | 0.599 | 0.548 | 0.577 | 0.602 | 0.578 | 0.709 |

*Figure 8.* Improvement of $\ell_{bayes}$ across increasing latent entropy on Machine Translation.

*Figure 9.* Improvement of $\ell_{bayes}$ across increasing latent entropy on multi-answer QA (Simplex).

## A.4. Larger LLMs

We also study decoding and uncertainty estimation on a large backbone LLM, namely Qwen-2.5-72B (Qwen Team, 2025). In Table 7 we find that our approach also improves generation quality for this model, consistent with Table 2. Further, in Table 8, we report PRR for uncertainty quantification using Bayes Risk versus other methods using the same Qwen-2.5-72B

*Table 5.* Uncertainty quantification PRR ↑ for different estimators on all tasks (best and runner-up). We measure how well the uncertainty aligns with metrics computed from the beam search response.

| Dataset | | MAQA $\{0,1\}^K$ | | TriviaQA $[K]$ | WMT19 $R^d$ | | CNN/DailyMail $\mathcal{G}$ | | | MAQA $\Delta^{K-1}$ |
|---|---|---|---|---|---|---|---|---|---|---|
| | | Hamming | F1 | Exact Match | COMET | Cosine | Hamming | F1 | Align Score | KL Divergence |
| Gemma-3-4B | $P$(True) | 0.148 | 0.134 | 0.343 | 0.082 | −0.036 | −0.182 | −0.207 | −0.031 | −0.147 |
| | SAR | 0.617 | 0.743 | 0.569 | **0.336** | *0.266* | 0.191 | **0.388** | 0.109 | *0.732* |
| | COCOA | 0.581 | 0.688 | 0.573 | 0.286 | 0.210 | *0.222* | 0.222 | 0.153 | 0.694 |
| | KLE | 0.675 | *0.749* | *0.697* | 0.319 | 0.161 | 0.099 | 0.044 | 0.041 | 0.697 |
| | SE | 0.776 | 0.719 | 0.666 | 0.146 | 0.079 | 0.174 | 0.199 | 0.050 | 0.703 |
| | MSP | 0.450 | 0.571 | 0.512 | 0.161 | 0.120 | 0.110 | 0.106 | 0.175 | 0.645 |
| | Latent Entropy | *0.804* | 0.705 | **0.713** | 0.232 | −0.006 | 0.137 | 0.170 | *0.240* | 0.682 |
| | $R_{\text{Bayes}\text{(ours)}}$ | **0.819** | **0.811** | 0.674 | **0.327** | **0.282** | **0.677** | **0.355** | **0.289** | **0.754** |
| Gemma-3-12B | $P$(True) | 0.420 | 0.383 | 0.250 | 0.151 | −0.031 | −0.186 | −0.031 | −0.096 | 0.063 |
| | SAR | 0.697 | *0.726* | **0.589** | *0.270* | *0.199* | 0.098 | *0.207* | *0.130* | *0.652* |
| | COCOA | 0.655 | 0.655 | 0.568 | 0.263 | 0.149 | 0.096 | 0.181 | 0.052 | 0.632 |
| | KLE | 0.670 | 0.686 | *0.582* | 0.236 | 0.097 | 0.079 | 0.016 | −0.012 | 0.614 |
| | SE | 0.721 | 0.672 | 0.566 | 0.081 | 0.035 | *0.199* | 0.140 | 0.023 | 0.620 |
| | MSP | 0.504 | 0.512 | 0.539 | 0.180 | 0.105 | 0.013 | 0.184 | −0.004 | 0.590 |
| | Latent Entropy | *0.767* | 0.678 | 0.577 | 0.220 | −0.066 | 0.151 | 0.177 | 0.119 | 0.613 |
| | $R_{\text{Bayes}\text{(ours)}}$ | **0.778** | **0.726** | 0.576 | **0.316** | **0.263** | **0.581** | **0.258** | **0.398** | **0.711** |
| Qwen3-4B | $P$(True) | 0.378 | 0.237 | 0.508 | 0.221 | **0.209** | 0.050 | 0.014 | 0.128 | 0.124 |
| | SAR | 0.464 | 0.590 | 0.519 | 0.235 | 0.177 | **0.299** | **0.200** | 0.140 | *0.516* |
| | COCOA | 0.398 | 0.569 | 0.516 | *0.276* | 0.199 | 0.231 | *0.252* | *0.153* | 0.498 |
| | KLE | 0.286 | 0.560 | 0.573 | 0.249 | 0.125 | 0.191 | 0.150 | 0.006 | 0.444 |
| | SE | 0.391 | *0.644* | 0.547 | 0.143 | 0.063 | 0.131 | 0.046 | 0.016 | 0.465 |
| | MSP | 0.328 | 0.483 | 0.510 | 0.200 | 0.184 | 0.070 | 0.187 | 0.084 | 0.498 |
| | Latent Entropy | *0.585* | 0.597 | *0.612* | 0.147 | −0.046 | 0.154 | 0.021 | 0.019 | 0.492 |
| | $R_{\text{Bayes}\text{(ours)}}$ | **0.776** | **0.788** | **0.631** | **0.309** | **0.272** | **0.487** | 0.152 | **0.272** | **0.569** |
| Qwen3-30B-A3B | $P$(True) | 0.024 | 0.042 | 0.261 | 0.026 | −0.014 | −0.070 | 0.013 | 0.001 | 0.180 |
| | SAR | 0.664 | *0.644* | 0.503 | 0.247 | 0.172 | *0.343* | **0.471** | 0.012 | *0.672* |
| | COCOA | 0.667 | 0.632 | **0.545** | 0.285 | 0.185 | 0.299 | 0.404 | 0.032 | 0.670 |
| | KLE | 0.610 | 0.613 | 0.506 | 0.278 | 0.117 | 0.217 | 0.322 | −0.115 | 0.627 |
| | SE | 0.648 | 0.629 | 0.500 | 0.063 | 0.048 | 0.225 | 0.344 | −0.078 | 0.652 |
| | MSP | 0.549 | 0.557 | *0.521* | 0.190 | 0.151 | 0.271 | 0.300 | *0.119* | 0.615 |
| | Latent Entropy | *0.701* | 0.617 | 0.483 | 0.157 | −0.062 | 0.115 | 0.174 | −0.128 | 0.630 |
| | $R_{\text{Bayes}\text{(ours)}}$ | **0.743** | **0.691** | 0.513 | **0.312** | **0.252** | **0.667** | 0.260 | **0.251** | **0.707** |

model. Again, we find that on most tasks with more complicated structure $\mathcal{L}$, our method yields high quality uncertainty estimates. Overall, Tables 7 and 8 verify that our method indeed also applies to larger LLMs and provides high quality generations and uncertainty estiamtes.

# B. Implementation Details

## B.1. Mapping Language Outputs to Latent Structures

Here, we detail how we realize the mappings $g_T$ to free-form text LLM responses into the corresponding latent structures. As our core assumption is that the LLM outputs free-form text that is embedded in the latent structure, i.e. the LLM is able to reasonable address the task we prompt it with, we discard answers that can not properly parsed through any mapping $g_T$. These occurrences are extremely rare as the instruction-tuned LLMs studied in this work respect the instruction prompts sufficiently well.

**Single-Answer Question Answering**   Similar to Kuhn et al. (2023), we use DeBERTa (He et al., 2021) as an entailment model to group the individual free-form text responses into semantic clusters that form the support of the domain over which the most frequent answer is selected.

**Multi-Answer Question Answering**   Given an answer string $s$, we want to extract all *semantically distinct* answers in that string. As an example to the question "Which ocean borders the USA? " the string $s =$" Both the Atlantic and Pacific Ocean border the USA. "gets mapped to {Atlantic, Pacific} by $g_T$. Concretely $g_T$ is implemented using gpt-4.1-mini with prompt 1.

*Table 6.* Uncertainty quantification AUC ↑ for different estimators on all tasks (best and runner-up). We measure how well the uncertainty aligns with metrics computed from the beam search response.

| Dataset | MAQA $\{0,1\}^K$ Hamming | F1 | TriviaQA $[K]$ Exact Match | WMT19 $R^d$ COMET | Cosine | CNN/DailyMail $\mathcal{G}$ Hamming | F1 | Align Score | MAQA $\Delta^{K-1}$ KL Divergence |
|---|---|---|---|---|---|---|---|---|---|
| **Gemma-3-4B** | | | | | | | | | |
| $P(\text{True})$ | 0.566 | 0.556 | 0.618 | 0.537 | 0.485 | 0.479 | 0.484 | 0.504 | 0.474 |
| SAR | 0.764 | 0.802 | 0.738 | 0.606 | 0.574 | 0.594 | 0.619 | 0.574 | 0.718 |
| COCOA | 0.747 | 0.773 | 0.777 | 0.589 | 0.555 | 0.577 | 0.569 | 0.589 | 0.704 |
| KLE | 0.793 | 0.802 | 0.771 | 0.608 | 0.559 | 0.539 | 0.510 | 0.510 | 0.705 |
| SE | 0.815 | 0.786 | 0.769 | 0.543 | 0.531 | 0.563 | 0.556 | 0.527 | 0.723 |
| MSP | 0.704 | 0.726 | 0.761 | 0.563 | 0.535 | 0.526 | 0.519 | 0.574 | 0.689 |
| Latent Entropy | 0.829 | 0.787 | 0.780 | 0.601 | 0.508 | 0.570 | 0.554 | 0.579 | 0.723 |
| $R_{\text{Bayes}}$(ours) | 0.835 | 0.828 | 0.772 | 0.608 | 0.584 | 0.745 | 0.617 | 0.607 | 0.757 |
| **Gemma-3-12B** | | | | | | | | | |
| $P(\text{True})$ | 0.673 | 0.657 | 0.645 | 0.557 | 0.495 | 0.432 | 0.458 | 0.474 | 0.534 |
| SAR | 0.788 | 0.785 | 0.775 | 0.587 | 0.562 | 0.570 | 0.576 | 0.542 | 0.715 |
| COCOA | 0.768 | 0.757 | 0.813 | 0.586 | 0.542 | 0.563 | 0.561 | 0.526 | 0.699 |
| KLE | 0.784 | 0.771 | 0.770 | 0.590 | 0.539 | 0.534 | 0.518 | 0.479 | 0.708 |
| SE | 0.783 | 0.762 | 0.763 | 0.518 | 0.513 | 0.572 | 0.534 | 0.501 | 0.718 |
| MSP | 0.724 | 0.715 | 0.804 | 0.572 | 0.532 | 0.523 | 0.543 | 0.515 | 0.678 |
| Latent Entropy | 0.814 | 0.781 | 0.767 | 0.600 | 0.493 | 0.557 | 0.551 | 0.559 | 0.715 |
| $R_{\text{Bayes}}$(ours) | 0.816 | 0.795 | 0.766 | 0.596 | 0.579 | 0.709 | 0.585 | 0.660 | 0.756 |
| **Qwen3-4B** | | | | | | | | | |
| $P(\text{True})$ | 0.595 | 0.569 | 0.675 | 0.523 | 0.550 | 0.517 | 0.483 | 0.539 | 0.540 |
| SAR | 0.735 | 0.733 | 0.717 | 0.580 | 0.549 | 0.604 | 0.563 | 0.526 | 0.662 |
| COCOA | 0.733 | 0.729 | 0.760 | 0.591 | 0.556 | 0.582 | 0.576 | 0.525 | 0.658 |
| KLE | 0.716 | 0.725 | 0.731 | 0.594 | 0.539 | 0.570 | 0.555 | 0.492 | 0.639 |
| SE | 0.731 | 0.747 | 0.706 | 0.533 | 0.517 | 0.577 | 0.548 | 0.504 | 0.639 |
| MSP | 0.707 | 0.698 | 0.757 | 0.573 | 0.553 | 0.546 | 0.550 | 0.520 | 0.655 |
| Latent Entropy | 0.770 | 0.733 | 0.724 | 0.584 | 0.502 | 0.613 | 0.554 | 0.531 | 0.639 |
| $R_{\text{Bayes}}$(ours) | 0.791 | 0.807 | 0.730 | 0.603 | 0.578 | 0.672 | 0.556 | 0.592 | 0.666 |
| **Qwen3-30B-A3B** | | | | | | | | | |
| $P(\text{True})$ | 0.493 | 0.506 | 0.636 | 0.517 | 0.506 | 0.515 | 0.514 | 0.499 | 0.541 |
| SAR | 0.764 | 0.748 | 0.732 | 0.583 | 0.548 | 0.625 | 0.657 | 0.510 | 0.716 |
| COCOA | 0.760 | 0.737 | 0.809 | 0.597 | 0.556 | 0.628 | 0.645 | 0.512 | 0.721 |
| KLE | 0.746 | 0.724 | 0.729 | 0.599 | 0.537 | 0.589 | 0.570 | 0.502 | 0.708 |
| SE | 0.754 | 0.729 | 0.723 | 0.517 | 0.512 | 0.576 | 0.573 | 0.475 | 0.720 |
| MSP | 0.729 | 0.712 | 0.801 | 0.577 | 0.547 | 0.605 | 0.592 | 0.517 | 0.690 |
| Latent Entropy | 0.774 | 0.736 | 0.726 | 0.588 | 0.499 | 0.586 | 0.590 | 0.507 | 0.719 |
| $R_{\text{Bayes}}$(ours) | 0.777 | 0.756 | 0.732 | 0.599 | 0.570 | 0.732 | 0.586 | 0.603 | 0.743 |

*Table 7.* Generation quality of structure-aware MBR (ours) versus other decoding strategies, including the sampled LLM response with minimal Bayes risk $\ell_{\text{Sample}}$ (best) using the Qwen-2.5-72B model.

| Dataset (Latent $\mathcal{L}$) | MAQA ($\{0,1\}^K$) Hamming ↓ | F1 ↑ | TriviaQA ($[K]$) Exact Match ↓ | WMT19 ($\mathbb{R}^d$) COMET ↑ | Cosine ↓ | Summarization ($\mathcal{G}$) Hamming ↓ | F1 ↑ | Align Score ↑ | MAQA ($\Delta^{K-1}$) KL ↓ |
|---|---|---|---|---|---|---|---|---|---|
| **Qwen-2.5-72B** | | | | | | | | | |
| Beam | 0.554 | 0.748 | 0.301 | 0.864 | 0.110 | 2.123 | 0.179 | 0.855 | 7.968 |
| Self-Consistency | 0.586 | 0.740 | 0.305 | 0.865 | 0.109 | 2.047 | 0.180 | 0.757 | 7.302 |
| AR | 0.729 | 0.740 | 0.308 | 0.864 | 0.109 | 2.094 | 0.197 | n.a. | 7.680 |
| $\ell_{\text{MAP}}$ | 0.521 | 0.747 | | 0.865 | 0.109 | 1.955 | 0.181 | 0.838 | 7.770 |
| $\ell_{\text{Sample}}$ | 0.495 | 0.752 | 0.308 | 0.863 | 0.109 | 1.866 | 0.196 | 0.862 | 8.191 |
| Bayes (ours) | 0.477 | 0.755 | | | 0.106 | 1.614 | 0.213 | 0.880 | 4.823 |

**Semantic Space Embedding**   We pass the LLM response $s \mid x$ into a semantic embedding space by directly feeding it into an EmbeddingGemma (Schechter Vera et al., 2025). To compute the COMET score, we need to map the embedding back to natural language. Since averaged embeddings do not have a clear free-form text correspondence, we instead use the embedding with the closest distance from the sampled responses.

**Knowledge Graphs**   We use KGGen (Mo et al., 2025) to extract knowledge graphs from text summaries $s \mid x$. The extracted relations $(u, e, v)$ between entities $u$ and $v$ are then used to induce the set of relevant entities and the graph to finally obtain $G = (V, E)$. We use clustering implemented by KGGen to group the entities and edges. Lastly, we compute the AlignScore corresponding to summary in form of a knowledge graph by using an inverse mapping $g_T^{-1}$ which simply joins all relations in a graph on fullstops to retrieve a summary back from the graph representation.

*Table 8.* Uncertainty quantification PRR ↑ for different estimators on all tasks (best and runner-up). We measure how well the uncertainty aligns with metrics computed from the Bayes optimal response. The backbone model is Qwen-2.5-72B.

| Dataset | | MAQA $\{0,1\}^K$ | | TriviaQA $[K]$ | WMT19 $R^d$ | | CNN/DailyMail $\mathcal{G}$ | | | MAQA $\Delta^{K-1}$ |
|---|---|---|---|---|---|---|---|---|---|---|
| | | Hamming | F1 | Exact Match | COMET | Cosine | Hamming | F1 | Align Score | KL Divergence |
| Qwen-2.5-72B | Latent Entropy | **0.769** | 0.682 | 0.487 | 0.181 | $-0.134$ | **0.046** | 0.114 | $-0.014$ | 0.390 |
| | $P$(True) | 0.329 | 0.319 | 0.388 | 0.210 | *0.127* | $-0.018$ | $-0.054$ | 0.003 | 0.189 |
| | SAR | 0.715 | *0.720* | 0.481 | 0.225 | 0.045 | $-0.050$ | 0.153 | $-0.093$ | 0.391 |
| | COCOA | 0.654 | 0.705 | **0.551** | *0.274* | 0.079 | $-0.011$ | 0.048 | *0.092* | **0.448** |
| | KLE | 0.669 | 0.687 | 0.479 | **0.301** | 0.068 | $-0.048$ | *0.198* | $-0.085$ | 0.393 |
| | SE | 0.641 | 0.619 | 0.480 | 0.244 | **0.167** | $-0.037$ | 0.105 | 0.019 | 0.382 |
| | MSP | 0.501 | 0.612 | *0.514* | 0.226 | 0.098 | $-0.022$ | $-0.016$ | **0.108** | 0.439 |
| | $R_{\text{Bayes(ours)}}$ | **0.785** | **0.742** | 0.475 | 0.255 | 0.096 | **0.223** | **0.330** | 0.047 | *0.446* |

*Table 9.* Examples of tasks, raw string generations, and induced latent representations. The mapping $g$ converts model outputs into structured latent spaces on which Bayes-optimal decoding is performed.

| Latent space $\mathcal{L}$ | Example question | Model output $y$ | Latent representation $\ell = g(y)$ |
|---|---|---|---|
| Classes $[K]$ | What is the capital of France? | Paris | `Paris` |
| Sets $\{0,1\}^K$ | Which countries border the USA? | Canada and Mexico. | $\{\underset{Mexico}{1}, 0, \underset{Canada}{1}, \underset{(.)}{0}\}$ |
| Real values $\mathbb{R}$ | What is the population of New York City (millions)? | Eight and a half million | 8.500.000 |
| Graphs $\mathcal{G} \cong \{0,1\}^E$ | What do you know about Paris? | Paris is the capital of France. [...] | $V : \{\text{Paris}, \text{France}, ...\} \quad E : \{\text{capital of}, ...\}$ |
| Probability simplex $\Delta^K$ | With what probability will it be sunny, cloudy, or rainy tomorrow? | With 80% probability it will be sunny tomorrow, with 15% cloudy, and with 5% rainy. | $[0.8, 0.15, 0.05]^T \in \Delta^2$ |
| Semantic embeddings $\{\ell \in \mathbb{R}^d : \|\ell\|_2 = 1\}$ | Istuntokauden uudelleenavaaminen | Resumption of the session | $[0.217, -0.345, 0.829, 0.673, ...]^T \in \mathbb{S}^d$ |

**Distributions (Probability Simplex)** We first prompt the LLM to output answers and probabilities in a JSON-like format (see Section E that can be parsed programatically. Afterwards, we again cluster the invididual answers using an entailment DeBERTa (He et al., 2021) and aggregate the corresponding probabilities.

## B.2. Bayes-optimal Action for Sets Under the $F_\beta$ Loss

Set-based responses can also be compared through the $F_\beta$ loss, which generalizes the $F_1$ loss to achieve different trade-offs between recall and precision. Intuitively, it treats outputting the correct response set $\ell$ as a binary classification problem over all $K$ classes. While there is no closed-form solution, Waegeman et al. (2014) propose an efficient algorithm to compute the Bayesian optimal action. The corresponding risk can be approximated using the definition from Equation (2) and Monte-Carlo sampling.

## B.3. Evaluation Metrics

We compute the Bayes-optimal action in closed-form from Equation (3) over $M = 20$ samples from the LLM embedded into the corresponding latent structure using $g_T$. For the Hamming-loss, to compare the hamming distance across sets of different support sizes, we always report the *normalized* Hamming loss as $d(\ell, \ell') = \frac{1}{K^*} \sum_{k=1}^K \mathbf{1}[\ell_i \neq \ell_i']$, where $K^* = \|\ell^*\|$ is the size of the true answer set. For the F1-score, we treat both the predicted set $\ell_{\text{Bayes}}$ and the true set $\ell^*$ as datasets of binary instances (i.e. if an element is included in the set or not) and then compute the geometric mean of precision and recall. For knowledge graphs, the structural Hamming distance corresponds directly to the Hamming distance on the set of graph edges and is subject to the same normalization as the set-based latent structure. We compute the AlignScore (Zha et al., 2023) of a knowledge graph between the reassembled free-form text summaries and a true reference summary.

For uncertainty quantification, we use PRR at a maximum rejection threshold of 50%, following (Vashurin et al., 2025b) to avoid artificial inflation of the score. The concordance $AUC$ is an estimate of $\mathbb{P}(y_i > y_j \mid \hat{y}_i > \hat{y}_j)$. We follow (Therneau & Atkinson, 2024) and discard ties on $y$, i.e. $y_i = y_j$ and treat ties on the estimator, i.e. $y_i = y_j$ with a score of $\frac{1}{2}$. The value of the resulting score can be interpreted similarly to the traditional AUCROC, with 0.5 corresponding to random chance and 1 to perfect ranking ability.

### B.4. Uncertainty Estimators

We implement uncertainty estimators based on `lm-polygraph 0.5.0` using the default settings as described by (Vashurin et al., 2025a). For COCOA, we utilize the version that leverages maximum sentence probability, referred to as $\text{COCOA}_{\text{MSP}}$. Moreover, as both COCOA and MSP are sample-specific, they are computed from the beam search output. Naturally, all methods use the same sampling budget as our Bayes risk estimator with $M = 20$.

## C. Connection to Existing Uncertainty Quantification Measures

### C.1. Relating Wasserstein Distance to Decision-Theoretic Uncertainty

Smith et al. (2025) introduce a Bayesian decision-theoretic framework for decomposing predictive uncertainty into reducible and irreducible components. To relate this framework to our proposed quantities, let

$$R_p^{\text{Bayes}} := \mathbb{E}_{\ell\sim p}\big[d(\ell, \ell_p^*)\big], \qquad R_q^{\text{Bayes}} := \mathbb{E}_{\ell\sim q}\big[d(\ell, \ell_q^*)\big],$$

where $p$ is the model distribution and $q$ is the ground-truth distribution. We assume $q$ to be perfectly approximated by the model as $n \to \infty$, so i.e. $q = p_\infty$. Then according to Smith et al. (2025), *irreducible uncertainty (AU)* is defined as

$$\text{AU} := R_q^{\text{Bayes}}.$$

This is the minimum Bayes risk under the true distribution $q$. The *total uncertainty (TU)* is given by

$$\text{TU} := R_p^{\text{Bayes}},$$

corresponding to the minimum Bayes risk under the learned model distribution $p$. Finally, the *reducible uncertainty (EU)* is defined as

$$\text{EU} := \text{TU} - \text{AU} = R_p^{\text{Bayes}} - R_q^{\text{Bayes}}.$$

It may assume negative values. We now relate these quantities to our uncertainty measure $W_1(p, q)$ and establish the following connection.

**Theorem C.1.** *Let $p$ be the model distribution and $q$ the true distribution on a metric space $(\mathcal{L}, d)$. Let*

$$R_p^{\text{Bayes}} := \mathbb{E}_{\ell\sim p}\big[d(\ell, \ell_p^*)\big], \qquad R_q^{\text{Bayes}} := \mathbb{E}_{\ell\sim q}\big[d(\ell, \ell_q^*)\big],$$

*where $\ell_p^*$ and $\ell_q^*$ denote Bayes–optimal actions under $p$ and $q$, respectively. Then*

$$\big| R_p^{\text{Bayes}} - R_q^{\text{Bayes}} \big| \ \leq \ W_1(p, q).$$

Under the interpretation of Smith et al. (2025), this result yields:

$$\big| R_p^{\text{Bayes}} - R_q^{\text{Bayes}} \big| = \big| \text{TU} - \text{AU} \big| = \big| \text{EU} \big| \ \leq \ W_1(p, q),$$

showing that the 1-Wasserstein distance upper-bounds epistemic uncertainty under the framework of Smith et al. (2025) and provides a natural geometric interpretation.

Importantly, this bound reveals an inherent asymmetry. While a small distributional distance implies low epistemic uncertainty—reflecting that the model closely matches the true distribution—a large distance does not necessarily entail high epistemic uncertainty. In particular, epistemic uncertainty may remain low when the model is *confidently wrong*. When the predictive distribution $p$ is highly concentrated but centered at a location different from that of $q$. In such cases, the model exhibits strong confidence despite substantial distributional mismatch.

### C.2. Connection to Bayesian Uncertainty Decomposition

We briefly relate our Bayes-risk view of uncertainty to the classical Bayesian decomposition of predictive uncertainty. In Bayesian uncertainty quantification, one assumes a posterior distribution $Q$ over model parameters $\Theta$. The predictive distribution is obtained by marginalizing over this posterior, $p(y) = \mathbb{E}_{\Theta\sim Q}\big[p(y \mid \Theta)\big]$. The (discrete) predictive entropy admits the standard decomposition (Depeweg et al., 2018)

$$\underbrace{H(Y)}_{\text{TU}} = \underbrace{\mathbb{E}_{\Theta\sim Q}\big[H(Y \mid \Theta)\big]}_{\text{AU}} + \underbrace{I(Y;\Theta)}_{\text{EU}}.$$

Equivalently, the epistemic term can be written as

$$I(Y; \Theta) = \mathbb{E}_{\Theta \sim Q}\Big[\text{KL}\big(p(Y \mid \Theta) \,\|\, p(Y)\big)\Big].$$

It therefore measures the dispersion of posterior predictive distributions around their marginal prediction. We now show how ur framework recovers this epistemic term as a special case. To that end, let the latent space itself be the probability simplex, $\mathcal{L} = \Delta^{K-1}$, such that each latent object $\ell \in \mathcal{L}$ is a predictive distribution over $K$ outcomes. Using the KL divergence as the loss $d_T$ between latent objects, the corresponding Minimum Bayes Risk is exactly the Mutual Information Lemma 3.4. Thus, the Mutual-Information-based definition of epistemic uncertainty arises as the Bayes risk of a second-order distribution over predictive distributions when the loss is instantiated with the KL divergence. Importantly, when choosing $\mathcal{L}$ as the probability simplex with KL divergence as distance measure, we do not make any assumptions about the structure of the underlying $K$ classes. Specifically, we also make no assumptions about the similarity or distance of each of the $K$ categorical elements. KL is sensitive to changes in probability mass, but it does not encode any geometry or task-specific distance between the underlying $K$ outcomes. Hence, two posterior predictive distributions may be assigned a high KL-divergence-based distance even if their respective masses are dispersed between outcomes that are similar under some loss function over these outcomes.

To carry the idea of task-aware distances to this setting of having the latent space be the second-order space of probability distributions, one could replace the KL-divergence-based dispersion term by a discrepancy between predictive distributions that respects the geometry of the first-order space. One natural example would be the Wasserstein distance. Using the Wasserstein distance induced by some distance over outcomes preserves the Bayesian interpretation of epistemic uncertainty as disagreement among posterior predictive distributions while making this disagreement aware to the task's geometry. Defining uncertainty according to second-order and first-order risk analogously to the information-based definition inherits its strengths and weaknesses (Wimmer et al., 2023). We leave a full formal treatment of such second-order and geometry-aware uncertainty decompositions to future work.

# D. Proofs

## D.1. Proof of Lemma 3.1

**Lemma 3.1.** *For any given class $\ell \in [K]$, let $p_\ell = \Pr(g_T(s) = \ell)$ denote the relative frequency among the MC samples. Under exact-match loss $d_T(\hat{\ell}, \ell) = \mathbf{1}\{\hat{\ell} \neq \ell\}$, a Bayes-optimal action is any mode of $p$,*

$$\ell_{Bayes} = \arg\max_{k \in [K]} \Pr(\ell = k) = \arg\max_{k \in [K]} p_k.$$

*The corresponding Minimum Bayes Risk is*

$$R_p(\ell_{Bayes}) := \mathbb{E}_{\ell \sim p}\big[\mathbf{1}\{\ell_{Bayes} \neq \ell\}\big] = 1 - \max_{k \in [K]} p_k.$$

*Proof.* Fix any prediction $\hat{y} \in [K]$. By definition,

$$R_p(\hat{y}) := \mathbb{E}_{y \sim p}\big[\mathbf{1}\{\hat{y} \neq y\}\big] = \Pr(\hat{y} \neq y) = 1 - \Pr(y = \hat{y}) = 1 - p_{\hat{y}}.$$

Hence, minimizing $R_p(\hat{y})$ over $\hat{y} \in [K]$ is equivalent to maximizing $p_{\hat{y}}$. Therefore any maximizer $y_p^* \in \arg\max_{k \in [K]} p_k$ is Bayes-optimal, and its risk equals

$$R_p(y_p^*) = 1 - p_{y_p^*} = 1 - \max_{k \in [K]} p_k,$$

which completes the proof. $\qquad\square$

## D.2. Proof of Lemma 3.2

**Lemma 3.2.** *Let $\ell \in \{0, 1\}^K$ be distributed according to $p(\ell \mid x)$ and consider the Hamming loss $d_T(\ell, \ell') = \sum_k^K \mathbf{1}[\ell_k \neq \ell'_k]$. The Bayes-optimal prediction $\ell^{Bayes}$ is given component-wise by*

$$(\ell_{Bayes})_i = \mathbf{1}\big\{\Pr(\ell_i = 1) \geq \tfrac{1}{2}\big\}.$$

*The corresponding Minimum Bayes Risk is*

$$R(\ell_{Bayes}) = \sum_{k=1}^{K} \min\big\{ \Pr(\ell_i = 1), \Pr(\ell_i = 0) \big\} \ \leq \ \tfrac{K}{2}.$$

*Proof.* By linearity of expectation, the expected Hamming loss of a prediction $\tilde{z} \in \{0, 1\}^{|Y|}$ decomposes across coordinates:

$$\mathbb{E}_z[\ell(\tilde{z}, z)] = \sum_{i=1}^{|Y|} \mathbb{E}_z[\mathbf{1}\{\tilde{z}_i \neq z_i\}] = \sum_{i=1}^{|Y|} \Pr(\tilde{z}_i \neq z_i).$$

Thus, minimizing the expected loss reduces to minimizing each coordinate independently. Fix an index $i$. If $\tilde{z}_i = 1$, the expected loss is $\Pr(z_i = 0)$; if $\tilde{z}_i = 0$, the expected loss is $\Pr(z_i = 1)$. Hence, the optimal choice is

$$\tilde{z}_i = \begin{cases} 1, & \text{if } \Pr(z_i = 1) \geq \Pr(z_i = 0), \\ 0, & \text{otherwise,} \end{cases}$$

which is equivalent to the stated threshold rule.

Substituting this Bayes-optimal choice back into the risk expression yields

$$R(z^{\text{Bayes}}) = \sum_{i=1}^{|Y|} \min\big\{ \Pr(z_i = 1), \Pr(z_i = 0) \big\}.$$

Since each term is at most $1/2$, we obtain $R(z^{\text{Bayes}}) \leq |Y|/2$, which completes the proof. $\qquad\square$

### D.3. Proof of Lemma 3.3

**Lemma 3.3.** *Let $\ell \in \mathbb{R}^d : \|l\|_2 = 1$ be distributed according to $p(\ell \mid x)$ with mean $\mu := \mathbb{E}_{\ell \sim p}[\ell]$. Consider the loss $d(\ell, \ell') = 1 - \langle \ell, \ell' \rangle$. If $\mu \neq 0$, the Bayes-optimal action is*

$$\ell_{Bayes} \ = \ \arg\min_{\ell \in \mathcal{L}} \mathbb{E}_{\ell \sim p}[1 - \langle \ell, \ell' \rangle] \ = \ \frac{\mu}{\|\mu\|},$$

*and the corresponding Minimum Bayes Risk is*

$$R(\ell_{Bayes}) \ = \ \mathbb{E}_{\ell \sim p}[1 - \langle \ell, \ell_{Bayes} \rangle] \ = \ 1 - \|\mu\|.$$

*Proof.* We start from the definition of the Bayes action under cosine distance on the unit sphere:

$$\begin{aligned} \ell_{Bayes} &:= \arg\min_{\ell' \in \mathcal{S}} \mathbb{E}_{\ell \sim p}\big[1 - \langle \ell, \ell' \rangle\big] \\ &= \arg\max_{\ell' \in \mathcal{S}} \mathbb{E}_{\ell \sim p}\big[\langle \ell, \ell' \rangle\big] \\ &= \arg\max_{\ell' \in \mathcal{S}} \langle \mathbb{E}_{\ell \sim p}[\ell], \ell' \rangle. \end{aligned}$$

Let $\mu := \mathbb{E}_{\ell \sim p}[\ell]$. Then the objective is $\langle \mu, \ell' \rangle$. Writing $\mu = \|\mu\| \, \hat{\mu}$ with $\hat{\mu} := \mu / \|\mu\| \in \mathcal{S}$, we obtain

$$\langle \mu, \ell' \rangle = \|\mu\| \langle \hat{\mu}, \ell' \rangle.$$

Since $\hat{\mu} \in \mathcal{S}$ and $\ell' \in \mathcal{S}$, their inner product satisfies $\langle \hat{\mu}, \ell' \rangle \leq 1$, with equality achieved when $\ell' = \hat{\mu}$. Hence the maximizer is $\ell_{Bayes} = \hat{\mu} = \mu / \|\mu\|$.

For the Bayes risk, we plug $s_p^*$ into the definition:

$$\begin{aligned} R_p(\ell_{Bayes}) &:= \mathbb{E}_{\ell \sim p}\big[1 - \langle \ell, \ell_{Bayes} \rangle\big] \\ &= 1 - \langle \mathbb{E}_{\ell \sim p}[\ell], \ell_{Bayes} \rangle \\ &= 1 - \Big\langle \mu, \frac{\mu}{\|\mu\|} \Big\rangle \\ &= 1 - \|\mu\|. \end{aligned}$$

This completes the proof. $\qquad\square$

## D.4. Proof of Lemma 3.4

**Lemma 3.4.** *Let $\ell \in \Delta^{K-1}$ be distributed according to $p(\ell \mid x)$ and consider the KL divergence $d_T(\ell, \ell') = \sum_k \ell_k \log \frac{\ell_k}{\ell'_k}$. The Bayes-optimal prediction is the mean:*

$$\ell_k^{Bayes} = \mathbb{E}_\ell[\ell].$$

*The corresponding Minimum Bayes Risk is*

$$R(\ell_{Bayes}) = \mathbb{H}(\mathbb{E}_\ell[\ell]) - \mathbb{E}_\ell[\mathbb{H}(\ell)].$$

*Proof.* Expanding the KL divergence,

$$\mathbb{E}_\ell[d_T(\ell, \ell')] = \mathbb{E}_\ell\left[\sum_k \ell_k \log \ell_k\right] - \sum_k \mathbb{E}_\ell[\ell_k] \log \ell'_k.$$

The first term is constant in $\ell'$. Thus, the problem reduces to

$$\ell^{\text{Bayes}} = \arg \min_{\ell' \in \Delta^{K-1}} - \sum_{k=1}^K \mathbb{E}_\ell[\ell_k] \log \ell'_k.$$

Introducing a Lagrange multiplier $\lambda$ for $\sum_k \ell'_k = 1$, we consider

$$\mathcal{L}(\ell', \lambda) = - \sum_k \mathbb{E}_\ell[\ell_k] \log \ell'_k + \lambda\left(\sum_k \ell'_k - 1\right).$$

Setting derivatives to zero gives

$$\ell'_k = \frac{\mathbb{E}_\ell[\ell_k]}{\lambda}.$$

Enforcing the simplex constraint yields $\lambda = 1$, and therefore

$$\ell_k^{\text{Bayes}} = \mathbb{E}_\ell[\ell_k].$$

For the minimum Bayes risk, plug $\ell^{\text{Bayes}}$ into the expected KL:

$$\begin{aligned}
R(\ell^{\text{Bayes}}) &:= \mathbb{E}_\ell\left[d_T(\ell, \ell^{\text{Bayes}})\right] \\
&= \mathbb{E}_\ell\left[\sum_{k=1}^K \ell_k \log \ell_k\right] - \sum_{k=1}^K \mathbb{E}_\ell[\ell_k] \log \ell_k^{\text{Bayes}} \\
&= -\mathbb{E}_\ell[\mathbb{H}(\ell)] - \sum_{k=1}^K \mathbb{E}_\ell[\ell_k] \log \mathbb{E}_\ell[\ell_k] \\
&= \mathbb{H}(\mathbb{E}_\ell[\ell]) - \mathbb{E}_\ell[\mathbb{H}(\ell)]
\end{aligned}$$

where $\mathbb{H}$ is the Entropy function. $\qquad\square$

**Theorem C.1.** *Let $p$ be the model distribution and $q$ the true distribution on a metric space $(\mathcal{L}, d)$. Let*

$$R_p^{\text{Bayes}} := \mathbb{E}_{\ell \sim p}\left[d(\ell, \ell_p^*)\right], \qquad R_q^{\text{Bayes}} := \mathbb{E}_{\ell \sim q}\left[d(\ell, \ell_q^*)\right],$$

*where $\ell_p^*$ and $\ell_q^*$ denote Bayes–optimal actions under $p$ and $q$, respectively. Then*

$$\left| R_p^{\text{Bayes}} - R_q^{\text{Bayes}} \right| \leq W_1(p, q).$$

*Proof.* For any fixed $\tilde{\ell} \in \mathcal{L}$,

$$\left| R_p(\tilde{\ell}) - R_q(\tilde{\ell}) \right| = \left| \mathbb{E}_{\ell \sim p}[d(\ell, \tilde{\ell})] - \mathbb{E}_{\ell' \sim q}[d(\ell', \tilde{\ell})] \right| \tag{14}$$

$$= \left| \mathbb{E}_{(\ell, \ell') \sim \gamma} \left[ d(\ell, \tilde{\ell}) - d(\ell', \tilde{\ell}) \right] \right| \tag{15}$$

$$\leq \mathbb{E}_{(\ell, \ell') \sim \gamma} \left[ \left| d(\ell, \tilde{\ell}) - d(\ell', \tilde{\ell}) \right| \right] \tag{16}$$

$$\leq \mathbb{E}_{(\ell, \ell') \sim \gamma} \left[ d(\ell, \ell') \right], \tag{17}$$

where $\gamma$ is any coupling of $(p, q)$ and the last inequality uses the reverse triangle inequality. Taking the infimum over all couplings $\gamma$ yields

$$\left| R_p(\tilde{\ell}) - R_q(\tilde{\ell}) \right| \leq W_1(p, q).$$

Now observe that

$$R_p^{\text{Bayes}} \leq R_p(\tilde{\ell}) \leq W_1(p, q) + R_q(\tilde{\ell}).$$

Since this holds for all $\tilde{\ell}$, choose $\tilde{\ell} = \ell_q^*$ to obtain

$$R_p^{\text{Bayes}} \leq W_1(p, q) + R_q^{\text{Bayes}}.$$

Reversing the roles of $p$ and $q$ yields the reverse inequality and therefore

$$\left| R_p^{\text{Bayes}} - R_q^{\text{Bayes}} \right| \leq W_1(p, q).$$

$\square$

# E. Prompts

*Prompt 1. $g_T$ Prompt for Set-Case*

```
You extract explicit, semantically distinct atomic answers from a model answer.

Rules:
- Extract only what the answer explicitly states; no inference.
- Split coordinated lists (commas, and/or, slashes) into separate items.
- Merge synonyms/rephrasings; keep one canonical surface form, preserving key head nouns.
- Remove leading articles/hedges; remove trailing punctuation/parentheticals unless part
    of the name.
- Ignore explanations/justifications; include quantities only if the quantity itself is
    the item.
- If no explicit item appears (I don't know, etc.), output: I don't know.
- Output strictly: one item per line, no bullets or extra text.

Follow the output format in the examples.

Example 1
Q: Which oceans border the USA?
A: The Atlantic and Pacific Oceans.
Output:
Atlantic Ocean
Pacific Ocean

Example 2
Q: What are the official languages of Switzerland?
A: German, French, Italian, and Romansh are the four official languages.
Output:
German
French
Italian
Romansh
```

```
Q: {question}
A: {answer}

Task: List all explicit, distinct answers from the model answer, one per line.
```

*Prompt 2.* $g_T$ Prompt for Simplex-Case

```
You extract explicit, semantically distinct atomic answers and their stated probabilities.

Rules:
- Extract only what is explicitly stated; no inference.
- Each atomic answer must have the probability stated in the text.
- Split lists only if items have separate probabilities.
- Merge synonyms/rephrasings into one canonical answer; SUM their probabilities.
- Convert percentages/ratios to decimals in [0,1].
- Do NOT renormalize or modify probabilities.
- If no explicit answer appears, output exactly: {"items":[{"answer":"I don't know","p
  ":1.0}]}
- Output valid JSON only, no extra text.

Schema:
{"items":[{"answer":string,"p":number}]}

Example
Q: Which oceans border the USA?
A: Atlantic (55%), Pacific (45%).
Output:
{{"items":[{{"answer":"Atlantic,"p":0.55}},{{"answer":"Pacific,"p":0.45}}]}}

Q: {question}
A: {answer}

Task: Extract answer probability pairs.
```

*Prompt 3.* Prompt for WMT19

```
Translate the following sentence into english. Only return the translation without any
other text.
Sentence: {question}
Translation:
```

*Prompt 4.* Prompt for Single-Answer QA

```
{question}. Only provide the answer without explanation.
```

*Prompt 5.* Prompt for Simplex

```
Given a question with multiple possible answers provide the individual answers together
with their associated probabilities. Only return the individual answers and
probabilities without explanations and ensure that the probabilities sum to 1.0. Question:
{question}.
```

*Prompt 6.* Prompt for Summarization

```
Article: {article}
Summarize the above article in three simple sentences containing the main points. Only
    give back the summary without newlines or other formatting.
Summary:
```

