# OpenReview forum: "Task-Awareness Improves LLM Generations and Uncertainty"
_ICML.cc/2026/Conference — ICML 2026 regular_

### Official Review · Reviewer_N5iy · 2026-02-21

**Soundness:** 4
**Presentation:** 4
**Significance:** 3
**Originality:** 3
**Overall Recommendation:** 5
**Confidence:** 5

**Summary:**

This paper proposes a generalized framework for uncertainty quantification (and decoding strategy using UQ) of LLMs for tasks where latent structure/dist is possible. Their framework is inspired by existing works (such as semantic entropy), but different from existing works; it's a theoretically grounded and unified/generalized framework. They utilize their framework for many tasks such as single-label/multi label classification tasks, knowledge graphs, ambiguous questions, etc. They show the effectiveness of their approach over existing decoding strategies and UQ methods.

**Compliance With Llm Reviewing Policy:**

Affirmed.

**Key Questions For Authors:**

- Do you have any idea how we can apply this framework to a long from generations which contains multiple claims?

**Limitations:**

Yes. The authors did a good job of discussing the limitations.

**Strengths And Weaknesses:**

Strengths:
- The paper is technically sound.
- The presentation is great, the paper is well-written and easy to follow.
- The main contribution of the paper, I think, is creating a theoretical base for existing works such as semantic entropy, SAR, and others from a Bayes risk perspective, further creating a unifying framework for other tasks as well. This is a clean/solid contribution.
- The task richness in experiments is great.

Weaknesses:
- There is no major weakness, but the impact of the paper is kinda limited due to its nature/topic, which is fine though.
- The title can be changed to something more descriptive and interesting. With the current form, it looks too generic.

---

> ### Author Rebuttal · Authors · 2026-03-31
>
> Thank you for your review! We are happy that the conceptual contribution of our work is well received! You are exactly right that, from the perspective of UQ, we generalize many concepts that implicitly use semantic clusters as a latent space, e.g. semantic entropy. We also appreciate that you think it is theoretically well-grounded and backed by a broad empirical evaluation.
>
> W1: Can you elaborate what you mean by “limited topic”? We believe that our framework is quite general and can be applied to most tasks where one requires more than just the natural language response from the LLM. While not every task may admit an easy-to-use structure, our benchmark already indicates our approach's flexibility. Especially in terms of UQ, we believe that our work makes significant progress in moving from QA to a more general perspective.
>
> W2: This is an interesting suggestion and we are open to changing the title. What do you think about: “Task-Aware Minimum Bayes Risk Improves Decoding and Uncertainty Estimation in LLMs”? Are there any specifics that you think our title lacks?
>
> Q1: A simple latent structure again would be mapping to the space of “sets over claims”, where Hamming and F1 losses can be used, similar to summarization. Using precision as a distance instead could enable focus on factuality. There may even be more involved structures (e.g. syntax trees in code or directed graphs in reasoning) that require designing suitable distance metrics and approximate MBR algorithms. This goes beyond the scope of our work but highlights the potential of the perspective of task-awareness that we bring up. We now included a short discussion in our future work paragaraph.
>
> Again, we want to thank you for feedback and hope that we could adequately address all open points!

---

> > ### Author Rebuttal · Reviewer_N5iy · 2026-04-03
> >
> > Thanks to the authors for their reply. I like the new title more. My concern about the impact of the paper is not a critique of the paper itself, but UQ is a relatively niche area, which is why my comment is more like a justification for a score of 5.
> >
> > I maintain my score. Good luck

---

### Official Review · Reviewer_eYgJ · 2026-03-08

**Soundness:** 2
**Presentation:** 3
**Significance:** 2
**Originality:** 2
**Overall Recommendation:** 4
**Confidence:** 3

**Summary:**

This paper proposes a novel Bayesian method for handling LLM uncertainty by incorporating task-specific latent structures. The authors use this task-dependent knowledge to extract structured core information from sampled responses, calculate the Bayes Risk, and synthesize new optimal answers. The authors also provide concrete definitions of these latent structures and Bayes Risk calculations for several common tasks. Experimental results across diverse tasks and models demonstrate that the proposed method improves generation quality and reduces uncertainty.

**Compliance With Llm Reviewing Policy:**

Affirmed.

**Ethical Review Concerns:**

yes

**Final Justification:**

While I appreciate the authors' transparency regarding model overconfidence, the marginal results on TruthfulQA suggest that their core assumption breaks down in challenging tasks. Since high-confidence hallucinations dominate real-world scenarios for LLMs, the method's broader applicability is fundamentally constrained. Furthermore, given the demanded prior knowledge of full task structures before solving (which are also inaccessible in real scenarios), this work may potentially only serve as an impractical toy model. Thus, I maintain my score.

**Key Questions For Authors:**

Please refer to weakness:
1) Including additional decoding and generation baselines, beyond beam search and self-consistency, is required to make the performance evaluations more solid.

2) More challenging datasets that explicitly require strict uncertainty reduction, such as TruthfulQA, should be included to justify the method's effectiveness in real uncertainty-related scenarios.

3) More related discussions and evidence should be included to explain why the proposed method could lead to performance gains despite the LLM's innate tendency to make confident but erroneous predictions (so-called "over-confidence"), especially in more complex scenarios.

**Limitations:**

yes

**Strengths And Weaknesses:**

**Strengths:**

1) The paper proposes a novel, lightweight method for post-inference uncertainty handling.

2) The experiments cover a wide range of tasks, making the evaluation comprehensive.

3) The presentation of the paper is relatively clear.

**Weaknesses:**

1) The baseline methods compared against are relatively few. Other than beam search, self-consistency, and variations of the proposed method, additional baselines could be included to make the evaluation more persuasive.

2) The evaluated datasets are relatively simple and do not explicitly test scenarios like hallucination reduction. It remains unclear how the method would perform on datasets that more strictly require reducing uncertainty, such as TruthfulQA.

3) A decrease in uncertainty, or selecting the LLM's most confident (or most probable) answer, does not necessarily lead to a performance increase, as LLMs are prone to making confident errors, especially on more complex tasks. Therefore, further justification is required to explain why the proposed method could lead to a performance gain despite the potentially erroneous innate signals within LLMs.

---

> ### Author Rebuttal · Authors · 2026-03-31
>
> Thank you for your review. We appreciate you like the versatility and experimental breadth of our work and the presentation of our paper. We think your points raise very interesting discussions and improvements:
>
> W1,Q1: Thank you for this suggestion, and we added vanilla autoregressive sampling, greedy, contrastive search [1], and a prompting strategy requested by reviewer *2nCL* as additional decoding baselines (see Table 1, [figshare](https://figshare.com/s/0da354977c220454b197?file=63330325)). We still clearly outperform other methods. In general, we attempted to select the most promising and best-performing baselines available, and, in particular, evaluate against other approaches that utilize latent distance, like the MAP estimate in the latent structure or standard MBR using the latent distance without computing the closed-form optimizer. If you think that there should be any other important baselines to compare against, we would greatly appreciate any references so we can include them!
>
> W2, Q2: We are grateful for your pointer toward more challenging settings in which LLMs typically exhibit higher uncertainty. We added TruthfulQA and chose a classification-latent-structure with 0-1 loss for our evaluation (Table 4  (Decoding) [figshare](https://figshare.com/s/0da354977c220454b197?file=63330325), Table 5 UQ [figshare](https://figshare.com/s/0da354977c220454b197?file=63330325)). Note that for the well-studied QA tasks, the MBR action is to return the majority vote response and improvement to existing methods is less pronounced than for other task structures. While we believe this is a nice addition to our framework, we want to stress that we believe our evaluation should explore a breadth of tasks and not be fine-tuned to solve one specific, difficult task necessarily. Our main point is to recognize and leverage task structure *if it is present*, and not explicitly to deal with overconfidence in LLMs.
>
> W3, Q3: This point warrants some clarification, which we also added to our paper. Our approach does not attempt to and can not solve issues of model overconfidence in a sense where the LLM consistently outputs a wrong response. As our method (like other methods) operates on the model’s output distribution, we can only leverage uncertainty signal that is present here. A distribution collapsed to a single, wrong answer does not indicate any uncertainty. In fact, any method that operates on the model’s output distribution will struggle with this case. We clarified that we do not claim to address this issue in the revision as well.
>
> What our approach can achieve is to default to some “average” response in case the model’s output distribution has large task-dependent semantic variability, i.e. when the LLM outputs very different responses. In contrast to other (MBR and non-MBR) methods that select one of the output responses, we compute the closed-form risk minimzing response. This response, on average, incurs lower error and hence leads to better performance. This is precisely what we show with Figure 2: If the model is uncertain (in a distributional sense), the MBR response has lower risk / loss than defaulting to one of the (likely incorrect) methods. If you want to discuss this further, we are happy to.
>
> We hope we could clarify some important points and supplement suitable new experiments. Should there be any other benchmarks you think are valuable or remaining concerns we are happy for further discussions and hope that you consider increasing your score otherwise.
>
> **References:**
>
> [1]: Su, Yixuan, et al. "A contrastive framework for neural text generation." *Advances in Neural Information Processing Systems* 35 (2022): 21548-21561.
>
> [figshare](https://figshare.com/s/0da354977c220454b197?file=63330325)

---

> > ### Author Rebuttal · Reviewer_eYgJ · 2026-04-02
> >
> > Thanks for the supplemented baselines. While I appreciate the authors' transparency regarding model overconfidence, the marginal results on TruthfulQA suggest that their core assumption breaks down in challenging tasks. Since high-confidence hallucinations dominate more practical and challenging scenarios, the method's broader applicability is fundamentally constrained. Thus, I maintain my score.

---

> > > ### Author Response · Authors · 2026-04-04
> > >
> > > Thank you for responding to the rebuttal and for appreciating our transparency; we think this is vital to the scientific process.
> > >
> > > R1: *Marginal improvements in TruthfulQA suggest that their core assumption breaks down in challenging tasks*
> > >
> > > Certainly, our improvements on TruthfulQA are less pronounced compared to other tasks, which is however consistent with the other dataset that “only” admits a 1-0-loss based categorical latent structure (TriviaQA): Here, the MBR action is simply to output the majority response, and our framework does not leverage the distributional diversity in the answers particularly strongly. The true power of our framework lies in exploiting more involved task structures. It would be interesting to study a challenging benchmark that involves, e.g., hallucinations that can take on different structures. The smaller performance gap to the baselines is likely due to the simple structure [K] and not the hallucination-related challenges of TruthfulQA. We again want to stress that the main point of our paper is to exploit the task structure and not to explicitly handle overconfidence. That we observe performance improvements even on these datasets, even for the simplest structure, in our opinion, strongly suggests that our framework is not limited to only “easier” settings at all, but quite the opposite.
> > >
> > > R2: *High-confidence hallucinations dominate more practical and challenging scenarios*
> > >
> > > While there certainly are scenarios where high-confidence hallucinations dominate, recent work finds that these overconfident hallucinations where the outputs collapse to a point-mass-distribution are limited: [1] find that high certainty hallucinations ( + in which the model actually possesses the correct knowledge called CHOKE hallucinations) only *“comprises 0.3%-2.1% of all hallucinations”.* In fact, assuming high-confidence hallucinations to be the dominant case would imply that models behave almost like deterministic predictors in these relevant practical cases, which, across a wide range of the literature, is not the case. We acknowledge that collapsed distributions are a limitation of our framework, but we believe the literature does not suggest that this is a dominant failure case that renders our method ineffective in practice. However, it is certainly an important area of research to determine how to handle these cases in their own right.
> > >
> > > As we show in our paper (Figure 2), as long as hallucinations are accompanied by structural diversity in the LLM’s (latent) answers, our framework can empirically mitigate this problem by defaulting to low-risk “average” responses instead of outputting the hallucination. We hope that these arguments make a convincing case that our framework is applicable even in challenging settings that often induce hallucinations.
> > >
> > > References:
> > >
> > > [1]: Simhi, Adi, et al. "Trust me, I’m wrong: LLMs hallucinate with certainty despite knowing the answer." *arXiv preprint arXiv:2502.12964* (2025).

---

### Official Review · Reviewer_GSF2 · 2026-03-12

**Soundness:** 3
**Presentation:** 4
**Significance:** 3
**Originality:** 4
**Overall Recommendation:** 5
**Confidence:** 4

**Summary:**

This paper addresses the need to model LLM uncertainty beyond the token space and proposes a structured latent space specifically for benchmarks which admit sensible structure, from single answer Q&A and multiple-answer Q&A to knowledge graphs and probability simplexes. It constructs adequate measures to model Minimum Bayesian Risk with decision-theoretic motivations. The paper showcases consistently strong results in the case of ambiguous multiple-answer Q&A modeled as probability simplexes and outperforms standard baselines such as Semantic Entropy.

**Compliance With Llm Reviewing Policy:**

Affirmed.

**Final Justification:**

Raising my score accordingly as the authors have shown appropriate additional baselines, discussed the matter of MBR as UQ and provided toy experiments which situate the paper better in existing UQ literature.

I would note that at this time the paper PDF in OpenReview is still the initial submission version. Ideally the authors can include their additional tables and figures in the camera-ready version.

**Key Questions For Authors:**

- Could the authors provide a rebuttal on the topic of MBR conflating risk with uncertainty?

- Could the authors please clarify how their work situates with Wimmer et al. [1] that was not mentioned in Related works? Also would the authors be willing to devise a controlled toy experiment to see if the proposed measures are well-behaved?

**Limitations:**

yes

**Strengths And Weaknesses:**

Strengths:

- Clear, detailed and task-aware formulations of MBR measures with benchmark-specific adaptations.
- Well-articulated manuscript.
- Strong and consistent UQ AUC results for the MAQA* dataset with probability simplexes.
- Overall I appreciate the direction of inducing structure into previously unstructured LLM tasks since uncertainty estimation methods struggle when limited to the unstructured natural language space.


Weaknesses:

- As pointed out in the Introduction section and in the Limitations section, the idea of inducing structure in admissible problem spaces is valuable, but it does limit applicability to other settings where the structure is much more complex than for Q&A (coding, multi-step logical reasoning, mathematics to name a few) or settings without inherent structure where it is difficult to achieve ground truth consensus (human preferences, graphical or musical art generation, generic chatbots to name a few).

- I maintain that Minimum Bayes Risk (MBR) is fundamentally a decision-theoretic concept related to optimal decision-making, rather than a direct measure of uncertainty. However, the authors appear to conflate decision-theoretic risk with uncertainty quantification, which led me to assign a lower score for Soundness. I expected the paper to incorporate established uncertainty measures, such as entropy or posterior variance. Entropy is employed only in Section 3.5 ("Ambiguous Ground-Truth: Probability Simplex"), where ground-truth probabilities are derived from LLM-expressed uncertainty - incidentally an approach that the authors themselves acknowledge as a weak proxy in Section 5 ("Related Work: UQ in LLMs"). Notably, Tables 4 and 6 show that entropy-based MBR applied to the MAQA* probability simplex data consistently outperforms all other baselines in terms of AUC. In contrast, non-entropy-based MBR methods applied to datasets such as TriviaQA, WMT19, and CNN/DailyMail underperform relative to other evaluated baselines. I find this contrast particularly compelling, as it raises the question of whether decision-theoretic MBR alone is sufficient or even appropriate for quantifying uncertainty.

- For the proposed structured latent space and associated MBR-based measures, I encourage the authors to more explicitly situate their work within the existing literature, such as Wimmer et al. [1], particularly when extending MBR to claims about the decomposability of aleatoric and epistemic uncertainty. Additionally, I would welcome empirical or theoretical evidence demonstrating how well-behaved these MBR measures are in simple, controlled, and intuitively interpretable uncertainty settings. Essentially, a toy example using the proposed measures would be welcome.

[1] Wimmer, Lisa, et al. “Quantifying Aleatoric and Epistemic Uncertainty in Machine Learning: Are Conditional Entropy and Mutual Information Appropriate Measures?” arXiv, 2022, https://doi.org/10.48550/arXiv.2209.03302

---

> ### Author Rebuttal · Authors · 2026-03-31
>
> Thank you for your review! We appreciate you like the concept of our work and it’s clarity as well as the strong UQ performance. We think your points start a very valuable discussion:
>
> W1: Indeed, the efficacy of our method hinges on how well a task can be represented in a latent structure, which can be challenging in more open-ended problems. Other UQ methods focus on QA and are often more restrictive (see baselines). Nonetheless, we believe that utilising structure is valuable even in complex tasks, e.g. code generation (syntax trees) or reasoning (directed graphs). Our method likely doesn’t solve all possible problems, but our exemplary tasks show that many problems admit a simple and easy-to-work-with structure.
>
> W2, W3, Q1, Q2: We agree that our uncertainty lacks a bit of context: There are many notions of uncertainty: E.g., distributional similarity/distance to the training data, purely Bayesian modelling of beliefs, and even some work using Bayes risk (BR) (see Appendix C and [2]). MBR actually is conceptually similar to Bayesian uncertainty. This is apparent when disentangling EU and AU by modelling the LLM’s outputs be a 2nd-order distribution. Variance / MI-based estimators quantify EU as the dispersion of the 2nd-order distribution. Equivalently, Bayes risk measures the expected distance from the distribution’s barycenter. It, too, captures variation, but also accounts for task-specific distances (see Figure 3c). Using KL as a latent distance precisely recovers MI (Table 1) as BR. We now also include across all tasks the latent entropy as a UQ baseline: we improve over this entropy through distance-awareness, see (Table 3, [figshare](https://figshare.com/s/0da354977c220454b197?file=63330325)).
>
> Other than Bayesian UQ, BR distinguishes between “beliefs” with the distance metric. Neither BR nor information-theoretic Bayesian UQ (e.g. MI) is strictly preferable in all settings. Leveraging task-specific distances correlates uncertainty better with performance: For BR, disagreeing beliefs with low distance (i.e. similar performance) induce less uncertainty. However, this may well come at the cost of decorrelating uncertainty with other objectives.
>
> When disentangling uncertainty by having the latent space be first-order distributions, we can define EU, AU and TU analogously to the information-theoretic Bayesian decomposition:
> - EU is the MBR w.r.t. the induced Wasserstein distance: $EU := MBR^2(Q) = \min_{p \in P(\mathcal{X})} \mathbb{E}_{q \sim Q} \left[W^1_d(q, p)\right]$
> - AU : average 1st-order risk: $AU := \mathbb{E}_{q \sim Q} \left[MBR^1(q)\right]$
> - non-additive TU as the Bayes risk of expected 1st-order distribution $\bar{q} = \mathbb{E}_{q \sim Q}[q]$, i.e. $TU := MBR^1( \bar{q} )$.
>
> Here, $Q$ is the LLM’s 2nd-order distribution, $d$ the task metric and $\mathcal{X}$ the 0-order space and the first-order MBR is $ MBR^1(q) = \min_{z \in \mathcal{X} } \mathbb{E}_{y \sim q}[ d(y, z) ]$.
>
> Note that in contrast to [2] and App. C, these definitions are estimates of “ground-truth” uncertainty. BR uncertainty increases when disagreeing beliefs have high 1st-order-distance: When using L1-distance over a discrete set of reals $\mathcal{X} = {1,2,10}$, uncertainty between 1 with 2 induces less overall uncertainty than disagreement between 1 with 10 (see Figure 1, [figshare](https://figshare.com/s/0da354977c220454b197?file=63330325), compare 4th to 5th column, bottom row). Bayesian UQ makes no distinction (middle row, compare 4th to 5th column).
> The 2nd-order distributions in our toy example show that for both losses, the proposed measures behave intuitively and similar to MI. Therefore, BR violates the same axioms proposed in [1] as MI. Please note that we do not claim MBR risk to solve any of MI’s issues. It is more a distance-aware generalization thereof.
> We added this discussion in greater detail to our appendix, including toy experiments and the formal definitions, and contextualized our method accordingly w.r.t. to [1]. We believe this is a very valuable addition to our paper!
>
> Again, we are very grateful for raising this interesting dicussion. We hope we could contextualize our work and risk-based UQ within existing literature better and both formally and intuitively justify using BR for UQ (disentanglement). We also hope the toy experiment clarified how BR can be used to disentangle EU and AU. We are open to discuss any remaining open points further! Otherwise, we hope that you consider increasing your score in support of accepting our paper!
>
> **References**:
>
> [1]: Wimmer, Lisa, et al. “Quantifying Aleatoric and Epistemic Uncertainty in Machine Learning: Are Conditional Entropy and Mutual Information Appropriate Measures?” arXiv, 2022
>
> [2]: Smith, Freddie Bickford, et al. "Rethinking aleatoric and epistemic uncertainty." *arXiv preprint arXiv:2412.20892* (2024).
>
> [figshare](https://figshare.com/s/0da354977c220454b197?file=63330325)

---

> > ### Author Rebuttal · Reviewer_GSF2 · 2026-04-04
> >
> > Raising my score accordingly as the authors have shown appropriate additional baselines, discussed the matter of MBR as UQ and provided toy experiments which situate the paper better in existing UQ literature.
> >
> > I would note that at this time the paper PDF in OpenReview is still the initial submission version. Ideally the authors can include their additional tables and figures in the camera-ready version.

---

### Official Review · Reviewer_2nCL · 2026-03-12

**Soundness:** 3
**Presentation:** 3
**Significance:** 3
**Originality:** 3
**Overall Recommendation:** 4
**Confidence:** 3

**Summary:**

Uses a hand-defined latent structure and distance measure to choose actions to minimize expected risk.  Gives example structures and distance measures for various data types.  Runs some experiments against other exotic decoding methods, but not the standard autoregressive usage method.

**Compliance With Llm Reviewing Policy:**

Affirmed.

**Key Questions For Authors:**

1) "We define the true uncertainty as the discrepancy between this predictive belief and the (unknown) true distribution p∗(ℓ | x) over latents"  What's wrong with entropy of predictive belief as a measure of uncertainty?
2) Do any of the methods in Table 2 simply use standard autoregressive sampling?  Seems like the most standard usage should be one of the baselines.
3) "existing MBR approaches cannot do better than selecting the LLM generation associated with the lowest risk".  Well, what if the distance metric is bad?  Surely on the original loss, other procedures could do better in principle?

**Limitations:**

yes

**Strengths And Weaknesses:**

Strengths:
 - They give real examples of their methods
 - They try on a few different LLMs

Weaknesses:
 1) Why do we need fancy decoding at all in autoregressive models?  This was never motivated.  That's where all the SOTA results are for these kinds of queries.  If you want to do something fancier, why not choose a setting like bio where you might need more exotic conditioning or generation order?
2) As far as I can tell, any use of this method needs a pre-defined ontology in the latent space and an encoder.  For free-form text modeling, this seems like a major limitation.
3) "This shows that the Bayes-optimal action outperforms other decoding methods, particularly when the LLM is uncertain about its response."  You can also ask an LLM explicitly to output the risk-minimizing response.  Seems like a natural baseline.
4) The models evaluated on are pretty small.
5) the method seems very niche, and the distance metrics have lots of degrees of freedom that it's not clear how to set.
6) this all seems so far from the SOTA, even amongst open-source models.
7) the paper discusses uncertainty held by the LLMs, but it's only talking about token-level uncertainty.  Modern LLMs can also sometimes explicitly reason about their own uncertainty (even if their output distribution is deterministic)

---

> ### Author Rebuttal · Authors · 2026-03-31
>
> Thank you for your review! We want to clarify with additional experiments, incl. AR decoding, that we provide in [figshare](https://figshare.com/s/0da354977c220454b197), Tables 1,2 that our work does not aim at replacing AR decoding as a default. Instead, if one uses an LLM for a specific task, our method enables better results than just using free-form AR LLM by doing “post-processing” through MBR.
>
> W1,Q2: All our baselines are based-on standard autoregressive (AR) sampling. You are right; we did not compare against simple AR: we now added it to our experiments (Tab. 1, figshare): We consistently outperform AR in all tasks and metrics. Are there any other SOTA methods you want us to compare against?
>
> Moreover, decoding methods like MBR bring improvements in many scenarios and do not require exotic settings. This is evidenced by the growing number of works applying MBR across diverse settings (see our related work section).
>
> W2: We think there is a misunderstanding here: We let the LLM generate text answers first (using AR); then, we map them to a latent space and obtain the MBR action. We agree that not every single task in LLMs may be suitable for this, as discussed in our limitations. But, even longer texts could be, for example, decomposed into claims. These can be mapped to a latent space of “sets of claims” [2], again enabling our framework. While not every task may be equally suitable for this, if it admits any structure, we demonstrate that our framework can leverage this effectively. Many problems addressed with LLMs admit structure (see our experiments): Here, we have strong performance gains over baselines.
>
> W3: Thank you for the nice suggestion. We added prompting the LLM to produce MBR outputs as a baseline (Tab. 1, figshare) and still outperform this.
>
> W4: We now include results for Qwen2.5-72B (Tab. 2, figshare).
>
> W5: We respectfully disagree that decoding and UQ in LLMs is a nieche topic, as evidenced by our related work section. The idea of MBR is already well-established in the LLM community (related work). We believe that the strong performance of our method also justifies our doing it.
>
> We are a bit unsure what you mean by “degrees of freedom”: All of our distances have no “hyperparameters” or such. You are however right that our framework requires selecting a distance metric for each task. However, almost every task admits some performance metric, e.g. accuracy (0-1) in QA. The power of our framework is specifically that it can be applied to *any* metric, as acknowledged by reviewers **eYgJ,N5iy.**
>
> W6: To our best knowledge, we compare to the most relevant baselines (now including AR and 3 others): we outperform them consistently. If you think that we still miss important baselines or benchmarks, could you please refer us to them or any other SOTA numbers for comparison? Otherwise, can you maybe specify what you mean by “seems far from SOTA”? Our evaluation shows that there is currently no other method, also not AR, that we do not outperform.
>
> W7: Our method doesn’t use token-level uncertainty but instead the (latent) output distribution over sentences. Indeed, there is work about LLMs verbalizing their uncertainty: we outperform this approach (P(True)) in our experiments. Can you please provide us with the reference to any baseline you think we are missing?
>
> Q1: Good idea! MBR generalizes latent entropy (see Figure 3c). To explicitly show this in the revision, we now also include latent entropy as a baseline: MBR consistently outperforms it (Tab. 3, figshare).
>
> Q2: see W1
>
> Q3: This seems to be a misunderstanding: The MBR answer is not necessarily the best response that can be achieved in general. We say that other MBR frameworks that do not compute the closed-form MBR response can only select the LLM’s answer with the lowest risk. This, by definition, will always induce higher risk than our closed-form MBR solution. We clarified this in our revision.
>
> What do you mean by “original” loss? Our work aims at improving performance in a dedicated task in terms of the task-metric, and MBR improves responses w.r.t. this metric. Can you give a reference other methods we should compare against?
>
> We hope that we could address some misunderstandings. Your feedback greatly helped us clarify the main points of our paper in the revision. We hope that by adding more decoding baselines (incl. AR) we show that our work is not an exotic replacement for standard generation but an add-on on for leveraging task-knowledge to augment AR generations. If you nevertheless believe we do not sufficiently demonstrate that we are SOTA, we would be happy if you could point out exactly which baselines / experiments we are missing to make our claims convincing. Otherwise, we kindly ask you to consider raising your score, particularly given that the other reviewers have no reservations about the experimental rigor of our work.
>
> **References**
>
> 1: https://arxiv.org/abs/2505.20295
>
> 2: https://arxiv.org/abs/2410.13246

---

> > ### Author Rebuttal · Reviewer_2nCL · 2026-03-31
> >
> > Thanks for the detailed response and extra results.  I did indeed misunderstand this method as an alternative to AR as opposed to an augment.  I really like that you included the suggested simpler baselines and somewhat larger models.
> >
> > I'm raising my score accordingly.

---

### Decision · Program_Chairs · 2026-04-30

**Decision:**

Accept (regular)

**Comment:**

LLM decoding and uncertainty estimation typically operate on raw text, ignoring the structure of the task. This paper observes that many tasks have a known output structure — sets of answers, graphs, embeddings, probability distributions — and that working in these spaces enables both better responses and more meaningful uncertainty. By mapping LLM samples into task-appropriate structured spaces and applying model based minimum risk (MBR) decoding, which aggregating samples to minimize average dissimilarity), the method synthesizes new answers rather than just selecting among samples. The authors use the induced Bayes risk as a task-aware uncertainty measure, generalizing semantic entropy.

Reviewers found the work to be well-motivated and well-validated (experiments consider QA, translation, summarization, and probabilistic prediction tasks on multiple LLMs) [GSF2, N5iy, 2nCL]. The Bayes-optimal responses consistently outperform beam search, self-consistency, and standard text-space MBR in both generation quality and uncertainty calibration.

The main concerns were missing baselines, e.g., standard autoregressive decoding and prompt-based MBR were initially absent [2nCL, eYgJ]. The authors added AR, greedy, contrastive search, and prompting baselines, outperforming all baselines.  During the discussion period, the authors derived a formal epistemic-aleatoric decomposition, along with toy experiments, which provided an elegant framework for understanding how task-aware Bayes risk relates to standard Bayesian uncertainty—recovering mutual information as a special case and grounding the decision-theoretic quantity as a principled uncertainty measure. This resolved concerns raised by GSF2 (bumping paper from 5->6).

Post-rebuttal scores are 5, 6, 4, 5 (mean 5.0). I recommend accept.